# CTC1-STN1 terminates telomerase while STN1-TEN1 enables C-strand synthesis during telomere replication in colon cancer cells

Xuyang Feng[1], Shih-Jui Hsu[1], Anukana Bhattacharjee[1,3], Yongyao Wang[1,2], Jiajie Diao ᴼ[1] & Carolyn M. Price[1]

Telomerase elongates the telomeric G-strand to prevent telomere shortening through conventional DNA replication. However, synthesis of the complementary C-strand by DNA polymerase α is also required to maintain telomere length. Polymerase α cannot perform this role without the ssDNA binding complex CST (CTC1-STN1-TEN1). Here we describe the roles of individual CST subunits in telomerase regulation and G-overhang maturation in human colon cancer cells. We show that CTC1-STN1 limits telomerase action to prevent G-overhang overextension. $CTC1^{-/-}$ cells exhibit telomeric DNA damage and growth arrest due to overhang elongation whereas $TEN1^{-/-}$ cells do not. However, TEN1 is essential for C-strand synthesis and $TEN1^{-/-}$ cells exhibit progressive telomere shortening. DNA binding analysis indicates that CTC1-STN1 retains affinity for ssDNA but TEN1 stabilizes binding. We propose CTC1-STN1 binding is sufficient to terminate telomerase action but altered DNA binding dynamics renders CTC1-STN1 unable to properly engage polymerase α on the overhang for C-strand synthesis.

[1] Department of Cancer Biology, University of Cincinnati, Cincinnati, OH 45267, USA. [2] School of Life Science and Technology, Jiaotong University, Xi'an, Shaanxi 710049, China. [3]Present address: Department of Neurological Surgery, University of California San Francisco, Helen Diller Cancer Research Centre, 1450 3rd Street, San Francisco, CA 94158, USA. Correspondence and requests for materials should be addressed to C.M.P. (email: carolyn.price@uc.edu)

Telomeres harbor a series of proteins that protect the chromosome end and aid in its replication. In mammals, the six protein shelterin complex is the main source of telomere protection[1,2]. TRF1 and TRF2 bind the TTAGG-G•AATCCC repeats of the telomere duplex, POT1 binds the 3′ ssDNA extension on the G-rich strand (termed the G-overhang) while TPP1 dimerizes with POT1 and links it to TRF1/2 via TIN2. Together, TRF2 and POT1 prevent the DNA terminus from activating ATM and ATR-mediated damage signaling and unwanted repair reactions.

Telomere replication is a multistep process that has evolved to prevent the telomere shortening that would otherwise occur because DNA polymerase is unable to replicate the DNA 5′ end[3]. Telomerase is central to this process because it elongates the G-overhang through addition of TTAGGG repeats. However, other players are also required, including the ssDNA-binding trimeric complex CST (CTC1-STN1-TEN1) which participates in multiple aspects of telomere replication[4,5].

The duplex region of the telomere is replicated by the conventional replication machinery with assistance from CST, TRF1, and various helicases which help prevent replication fork stalling during passage through the repetitive G-rich sequence[5,6]. The DNA termini are then processed by nucleases to generate the 3′ overhang necessary for telomerase action[7,8]. Telomerase is aided by TPP1 which stabilizes telomerase association with the overhang and stimulates enzyme activity[9–11]. However, telomerase only extends the overhangs by ~60 nt and CST is thought to limit the amount of DNA that is added[12,13]. The final step in telomere replication occurs several hours later and involves synthesis of the complementary C-strand by DNA polymerase α-primase (pol α)[12]. This process, termed C-strand fill-in, converts the internal portion of the overhang into dsDNA. C-strand fill-in is absolutely required to prevent telomere shortening because the ssDNA generated by telomerase cannot be converted into dsDNA without this reaction (Supplementary Fig. 1a). CST is essential for C-strand fill-in[4] most likely because it enables pol α to engage correctly with the overhang in the absence of a replisome. In vitro studies indicate CST enhances pol α priming by stimulating the switch from RNA to DNA synthesis[14–16].

In addition to its telomeric roles, CST helps resolve replication problems throughout the genome[17,18]. The complex localizes preferentially to G-rich and repetitive elements where it prevents or resolves replication fork stalling[19]. It is likely that the role of CST in telomere duplex replication and genome-wide replication rescue are related and involve removal of DNA structures such as G-quadruplexes (G4)[20]. CST can also rescue stalled replication by facilitating firing of dormant replication origins[18]. Just how CST functions to resolve such a wide range of replication issues has been unclear. However, recent studies indicate that the answer lies in its structural similarity to Replication Protein A (RPA) the main eukaryotic ssDNA binding protein[20,21].

RPA, is a trimeric complex that is essential for DNA replication, repair, and recombination[22]. It functions by directing assembly/disassembly of complexes needed for these reactions and by melting unwanted DNA secondary structure. RPA binding is very dynamic because it contacts DNA through four OB folds, which can individually release and re-bind DNA without causing the entire complex to dissociate[23–25]. As a result, RPA can diffuse along DNA to melt secondary structure or displace bound proteins. Also, regions of ssDNA become exposed enabling protein loading. CST resembles RPA in that it harbors multiple OB-folds (one each in STN1 and TEN1, 5–6 predicted in CTC1)[21,26] and the structures of the small subunits are largely superimposable[27,28]. Recent studies indicate that CST also binds dynamically and this dynamic binding likely underlies the ability

of CST to melt G4 structure[20]. It may also provide a mechanism to engage partners such as pol α on ssDNA.

Despite the above similarities between CST and RPA, the overall architecture and functions of the two complexes are quite distinct[21]. Thus, it remains to be determined how the individual subunits of CST contribute to the various aspects of CST function. Here we dissect the contributions of individual subunits to specific steps in telomere replication and the overall dynamics of CST binding to ssDNA. We demonstrate that CTC1 and STN1 are sufficient to limit telomerase action and prevent telomeric DNA damage signaling but TEN1 stabilizes CTC1 and STN1 binding and is essential for C-strand fill-in and telomere length maintenance.

## Results

**Effect of human TEN1 disruption on telomere integrity.** To better understand how CST performs its various roles in telomere replication, we generated human HCT116 cells with a conditional TEN1 gene disruption so we could compare the effects of TEN1 and CTC1 loss[4]. CRISPR/Cas9 genome editing was used to introduce LoxP sites into the introns flanking exon 3 of the TEN1 locus (Fig. 1a). Cre recombinase was then introduced by infection with retrovirus encoding Cre-ER fusion protein. Individual clones were isolated, treated with tamoxifen to activate Cre, and screened for efficient gene disruption and loss of TEN1. Multiple clones were identified and one was chosen for in depth analysis (Fig. 1b–c, Supplementary Fig. 2i–j).

In preliminary experiments, we examined the effect of TEN1 removal on cell growth over a 3-week period (Fig. 1d, Supplementary Fig. 1b–d). $CTC1^{-/-}$ HCT116 cells almost cease proliferation within this timeframe[4], but surprisingly, growth of the $TEN1^{-/-}$ cells remained unchanged. As the stability of individual CST subunits can depend on expression of the other two subunits[15,29,30], we also examined whether TEN1 removal causes a decrease in STN1 or CTC1. Western blot analysis indicated partial loss of STN1 but CTC1 levels appeared unaffected (Fig. 1c).

To assess the effect of TEN1 deletion on telomere maintenance, we used fluorescence in situ hybridization (FISH) to examine telomere integrity. Metaphase spreads were prepared from TEN1 or CTC1 conditional cells after 0–14 days tamoxifen treatment and hybridized with a PNA probe to the telomeric G-strand (Supplementary Fig. 1e). The FISH revealed that $TEN1^{-/-}$ cells resembled $CTC1^{-/-}$ cells in that they showed an increase in chromosome ends lacking detectable FISH signal (5.5% signal free ends (SFE) by day 14) and a slight increase in sister chromatid associations (Supplementary Table 1). The level of telomere fusions was low (0.37%), although slightly higher than in the $CTC1^{-/-}$ cells (0.12%).

The decline in cell growth after CTC1 disruption is at least partially caused by an increase in telomeric DNA damage signaling[4]. However, the DNA damage signaling does not correlate with loss of telomeric dsDNA (i.e., decreased telomere FISH signal). Instead, it occurs at chromosome ends that retain telomeric DNA, but which have gained abnormally long overhangs. To assess whether TEN1 disruption also results in telomeric damage signaling, we looked for γH2AX at chromosome termini. Metaphase spreads were prepared from $CTC1^{-/-}$ and $TEN1^{-/-}$ cells by cytospin, γH2AX was detected by immunolocalization and telomeric DNA by FISH. The $CTC1^{-/-}$ cells showed the expected increase in telomeric γH2AX with most staining co-localizing with telomere FISH signal (Fig. 1e–f). However, the $TEN1^{-/-}$ cells showed no increase in γH2AX either at telomeres or elsewhere on the chromosome. The lack of damage signaling can explain the normal growth rate of $TEN1^{-/-}$

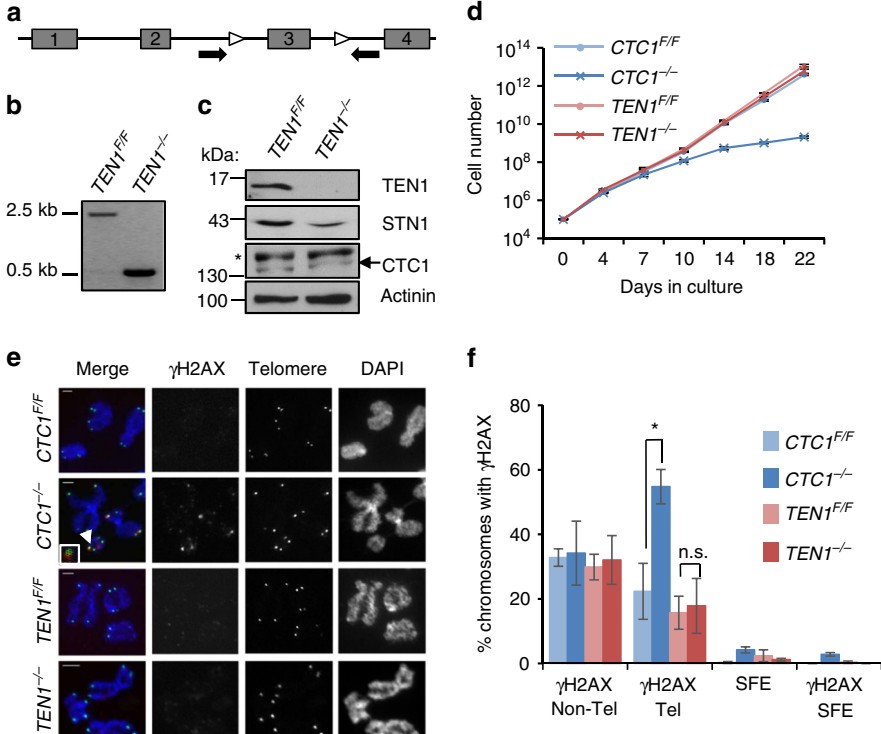

**Fig. 1** TEN1 gene disruption. **a** Modified *TEN1* gene locus. White arrowheads: loxP sites flanking exon 3. Black arrows: PCR primers used to verify exon deletion. **b** PCR verifying gene disruption. TEN1 conditional cells (*TEN1^F/F^* cells expressing Cre-ER) were grown with (*TEN1^−/−^*) or without tamoxifen (*TEN1^F/F^*) for 7 days. **c** Western blot analysis showing levels of TEN1, STN1, and CTC1 in same cells as **b**. Blot was probed with antibody to TEN1, CTC1, STN1, or actinin as a loading control. *non-specific band. **d** Representative growth curves showing proliferation after TEN1 or CTC1 disruption. **e**, **f** γH2AX localization on metaphase chromosomes after 12 days growth with/without tamoxifen. Scale bars = 2 μm. **e** Images showing γH2AX staining. Telomeres detected by FISH (green), γH2AX by immunostaining (red), chromosomes are counterstained with DAPI (blue). Arrow indicates FISH and γH2AX co-localization, insert shows enlargement of same region. **f** Quantification of chromosomes with γH2AX foci on chromosome arms (Non-Tel), at one or more telomeres with detectable telomeric DNA (Tel) or at signal free ends (SFE). *N* = 3 independent experiments, error bars indicate mean ± SEM *$P < 0.05$. ≥300 chromosomes scored per time point

cells. It also suggested that, unlike *CTC1^−/−^* cells, *TEN1^−/−^* cells may not gain extremely long G-overhangs. We note that both *CTC1^−/−^* and *TEN1^−/−^* cells lacked γH2AX staining at most of the chromosome ends with undetectable FISH signals (SFE). We surmise that these ends retain sufficient telomeric dsDNA to maintain end-protection.

**TEN1 deletion causes modest G-overhang elongation**. G-overhang length was examined by non-denaturing in-gel hybridization using DNA isolated from TEN1 or CTC1 conditional cells after various times of tamoxifen treatment. The DNA was separated briefly in agarose gels and hybridized with a probe to the telomeric G-strand under non-denaturing conditions (Fig. 2a, Supplementary Fig. 2k). The DNA was then denatured and re-hybridized with the same probe to provide a loading control. To distinguish G-overhang signal from internal regions of ssDNA, control samples were digested with Exonuclease1 (Exo1) prior to restriction digestion. The non-denaturing hybridization gave essentially no signal with the Exo1-digested DNA indicating that, similar to CTC1 loss[4], removal of TEN1 does not lead to accumulation of ssDNA within the telomere duplex. In contrast, the signal from the non Exo1-treated samples gradually increased with time after TEN1 deletion, indicating an increase in G-overhang length. However, quantification of the signal revealed that the magnitude of the increase was much smaller than that observed after CTC1 loss (Fig. 2b, Supplementary Figs. 2a–b, 9)[4].

Over 14 days, overhang signal increased by ∼1.8 fold in *TEN1^−/−^* cells compared with >4-fold in *CTC1^−/−^* cells.

The modest increase in overhang signal in *TEN1^−/−^* cells was similar to that observed after TEN1 or STN1 depletion with shRNA[5,30,31]. Since STN1 levels are affected by TEN1 loss (Fig. 1c)[30], it was possible that the small change in overhang length after TEN1 disruption was caused by a decrease in STN1 rather than a direct effect of TEN1 loss. To test this possibility, exogenous FLAG-STN1 or TEN1 were overexpressed in *TEN1^F/F^* cells (Supplementary Fig. 2c) to ensure a high level following endogenous TEN1 disruption. G-overhang length was then examined at various times after tamoxifen addition (Fig. 2c–d). The TEN1 overexpression prevented G-overhang elongation whereas the STN1 overexpression did not. Thus, loss of TEN1 directly affects G-overhang regulation.

Previous studies showed that the increase in G-overhang length after STN1 or TEN1 knockdown reflects a deficiency in C-strand synthesis[5,31]. To confirm the role of TEN1 in this process, we monitored cell-cycle related changes in overhang length. TEN1 conditional cells were synchronized in early S-phase with a double thymidine block, released into S-phase and harvested at intervals for G-overhang analysis (Supplementary Fig. 2d–f). Quantification of overhang signals indicated that *TEN1^F/F^* cells showed a slight increase in overhang abundance as they transitioned from early S-phase (0 h) into mid S-phase (6 h). The overhang signal then declined as the cells passed into G2 (9 h) and G1 of the next cell cycle (12 h). In *TEN1^−/−^* cells, the increase in overhang length followed a similar pattern, however, the decline during late S/G2

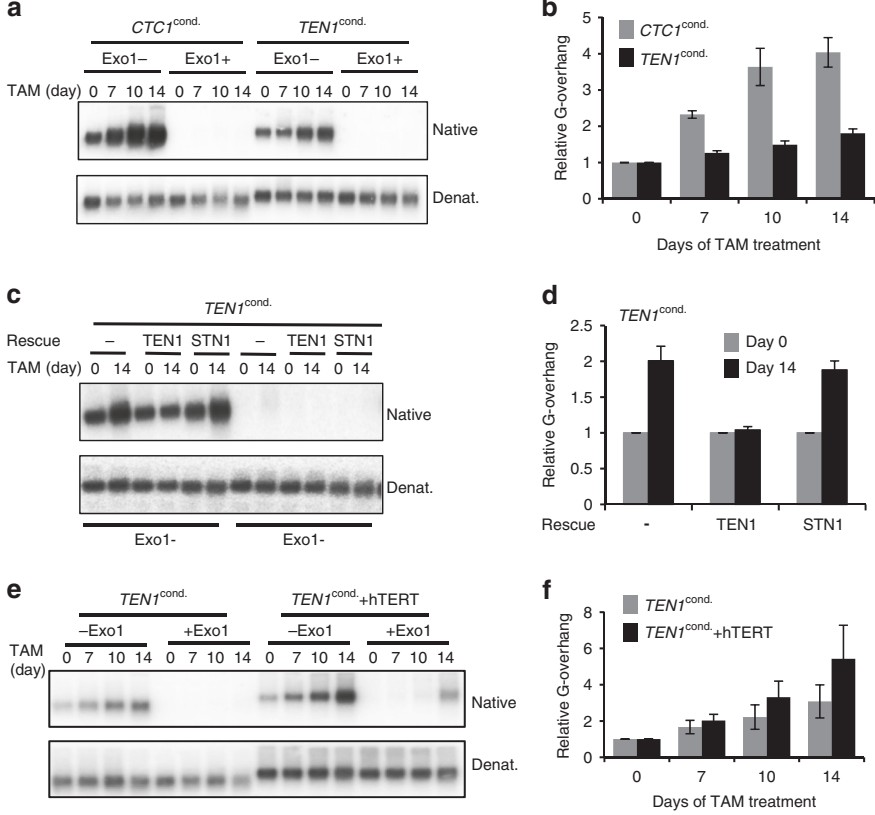

**Fig. 2** TEN1 loss causes modest G-overhang elongation. G-overhang abundance was analyzed by in-gel hybridization in the indicated cell lines following various times of tamoxifen (TAM) treatment. **a, b** TEN1 or CTC1 conditional cells. **c d** TEN1 conditional cells overexpressing exogenous TEN1 or STN1. **e f** TEN1 conditional cells with/without hTERT overexpression. **a, c, e** Gels showing hybridization of TAA($C_3TA_2$)$_3$ probe to genomic DNA under native and denaturing conditions. **b, d, f** Quantification of relative G-overhang abundance. Overhang abundance in $TEN1^{-/-}$, $CTC1^{-/-}$, and $TEN1^{-/-}$-hTERT cells was normalized to that of $TEN1^{F/F}$, $CTC1^{F/F}$, or $TEN1^{F/F}$-hTERT cells respectively (i.e., $t = 0$). Error bars indicate mean ± SEM, $n = 3$ independent experiments

was delayed and the overhangs remained longer in the next G1. The amount of initial overhang elongation was slightly less than that observed previously in HeLa cells[5], probably because the HCT116 cells synchronized in early S-phase rather than at the G1/S boundary. Nonetheless, the changes in overhang length were quite consistent. Importantly, the timing and extent of the delay in overhang shortening in the $TEN1^{-/-}$ cells was similar to that seen after STN1 or TEN1 knockdown[5,30,31]. The increase in overhang length during S-phase results from G-strand extension by telomerase combined with C-strand resection by nucleases[7,8,12]. Subsequent overhang shortening in late S/G2 occurs when a portion of the overhang is converted to dsDNA by C-strand fill-in[12]. Thus, the delay in overhang shortening in $TEN1^{-/-}$ cells confirms that TEN1 participates in the C-strand fill-in reaction.

**Removal of TEN1 causes net G- and C-strand shortening**. In $CTC1^{-/-}$ cells, the large increase in G-overhang length comes from a combination of C-strand shortening and G-strand elongation[4]. The C-strand shortening results from the lack of C-strand fill-in while the G-strand elongation is thought to occur because CST is no longer available to limit G-strand extension by telomerase[13]. Our finding that TEN1 disruption causes only modest overhang elongation suggested that TEN1 participates in C-strand fill-in but not telomerase regulation. To explore this possibility, we examined the effect of TEN1 disruption on telomere length.

In initial experiments, we used Southern hybridization to visualize terminal restriction fragments (TRFs). We simultaneously monitored telomerase activity. Although TEN1 removal had no effect on telomerase activity, average telomere length gradually

shortened by up to 1.0 kb over 14 days (Fig. 3a–b, Supplementary Fig. 2l–m, Supplementary Fig. 3a). This telomere shortening in the presence of abundant telomerase activity fits with TEN1 being required for C-strand fill-in (Supplementary Fig. 1a)[4]

Interestingly, the overall pattern of telomere shortening in $TEN1^{-/-}$ cells was different from that of $CTC1^{-/-}$ cells. Whereas $TEN1^{-/-}$ cells only exhibit telomere shortening, disruption of CTC1 initially leads to an apparent increase in TRF length (Fig. 3c, Supplementary Fig. 2m)[4]. This increase in length results from overextension of the telomeric G-strand. Telomere shortening due to loss of C-strand fill-in then becomes apparent at later time points. The lack of increase in TRF length after TEN1 disruption suggested that TEN1 plays no role in preventing G-strand overextension. To address this possibility, we used Q-FISH to directly examine G- and C-strand length. Metaphase spreads were prepared from $TEN1^{-/-}$ or $CTC1^{-/-}$ cells after various times after tamoxifen addition and hybridized with FISH probes to either the G- or the C-strand. Quantification of the FISH signal in $CTC1^{-/-}$ cells confirmed that the G-strands initially undergo a net elongation before gradually shortening, whereas the C-strands only undergo shortening Supplementary Fig. 3b). In the $TEN1^{-/-}$ cells, no net growth of the G-strands was visible (Fig. 3d–e). Instead, both the G- and the C-strands showed progressive shortening. We therefore conclude that TEN1 is not needed to prevent G-strand elongation.

**TEN1 is unnecessary for telomerase recruitment**. Although loss of C-strand fill-in results in gradual telomere shortening in telomerase expressing cells[4], this phenotype is also characteristic of a deficiency in telomerase recruitment or engagement at the

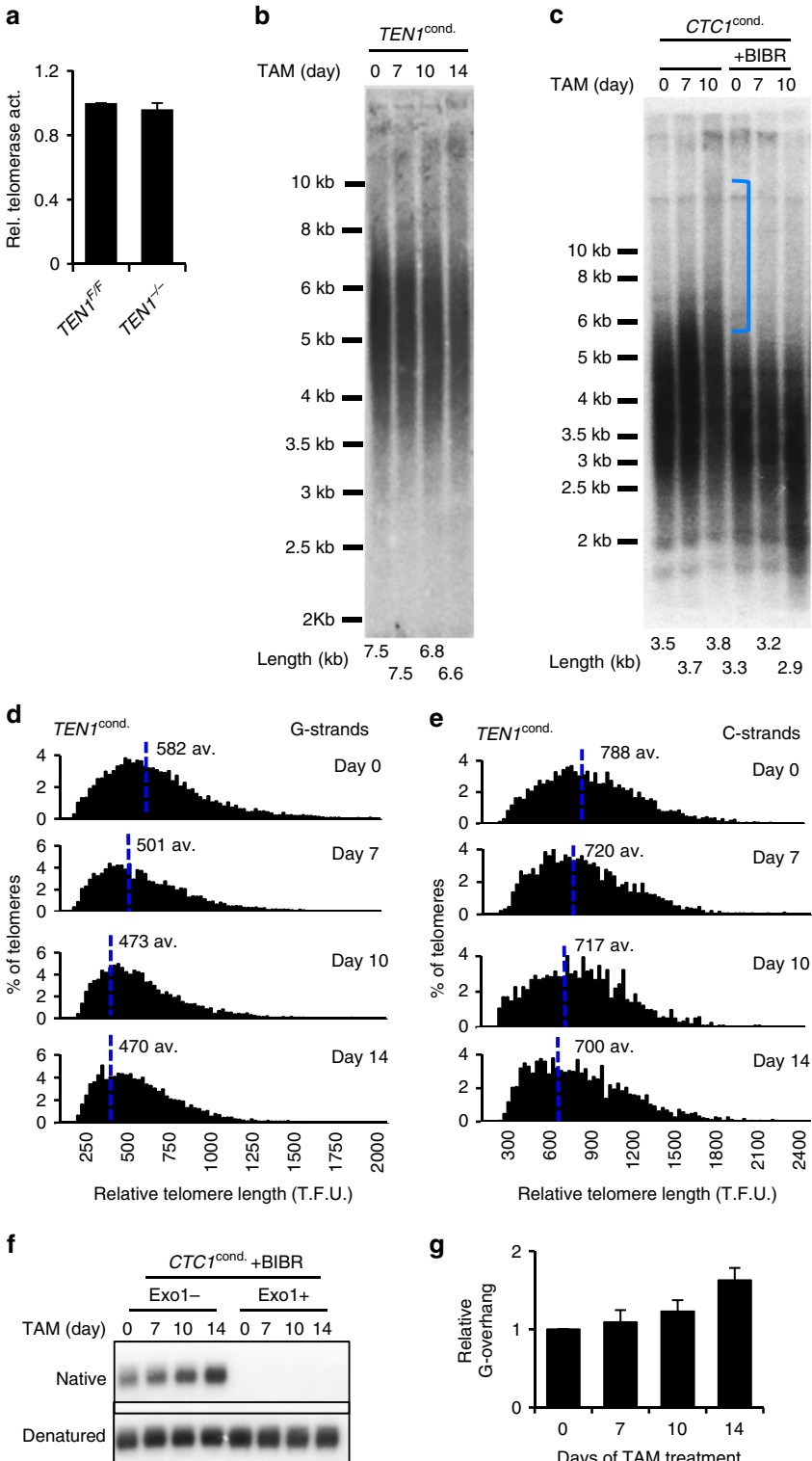

**Fig. 3** TEN1 disruption results in telomere shortening without G-strand overextension. **a** Quantification of TRAP assay showing telomerase activity in TEN1 conditional cells with/without 14 days tamoxifen treatment. **b c** Southern blots showing terminal restriction fragments in TEN1 (**b**) or CTC1 (**c**) conditional cells treated with tamoxifen for the indicated times. Mean telomere length is indicated below each lane. **c** Cells were grown with/or without BIBR1532 for 14 days. Bracket indicates telomere elongation. **d e** Analysis of telomere length by Q-FISH. Metaphase spreads were hybridized with $(C_3TA_2)_3$ G-strand probe (**d**) or $(G_3AT_2)_3$ C-strand probe (**e**). Histograms show distribution of relative telomere lengths expressed as fluorescence intensity (TFU telomere fluorescence unit). A minimum of 100 TFU was set as the cut-off. av.; median value, also shown by blue line. >2000 telomeres quantified per sample. **f g** G-overhang analysis in CTC1 conditional cells after BIBR1532 treatment. **f** Gel showing hybridization of $TAA(C_3TA_2)_3$ probe under native and denaturing conditions. **g** Quantification of G-overhang abundance. Signal from $CTC1^{-/-}$ cells was normalized to that of $CTC1^{F/F}$ (day 0) cells. Error bars indicate mean ± S.E.M., $n = 3$ independent experiments

telomere[3,9]. Thus, it remained possible that the telomere short-ening observed after TEN1 deletion reflected a dual requirement for TEN1 in C-strand fill-in and telomerase-mediated G-strand extension. We therefore set out to address whether TEN1 is needed for telomerase action.

Previous studies have shown that telomerase overexpression disrupts normal telomere length regulation and leads to telomere elongation[32]. However, no elongation will occur if telomerase is unable to gain access to the telomere[33]. Thus, to test whether TEN1 is required for telomerase action, we examined the effect of telomerase overexpression in $TEN1^{-/-}$ cells. Since the combined action of telomerase and C-strand fill-in are required for telomere growth (i.e., extension of the telomere duplex), we monitored the effect of telomerase overexpression on G-overhang length rather than overall telomere length. The rational was that if telomerase action is independent of TEN1, telomerase overexpression should cause additional G-overhang elongation in $TEN1^{-/-}$ cells. However, if TEN1 is required for telomerase action, G-overhang length would remain unchanged.

Telomerase overexpressing cells were generated by transfecting TEN1 conditional cells with an hTERT retroviral expression construct followed by hygromycin selection (Supplementary Fig. 2g). Telomere length analysis by Southern hybridization indicated revealed the expected telomere elongation indicating telomerase overexpression (Supplementary Fig. 2h). G-overhang length was then examined using cells grown with or without tamoxifen for various times (Fig. 2e–f). As before, $TEN1^{-/-}$ cells with normal levels of telomerase expression exhibited a fairly modest increase in overhang length. However, the TERT overexpression caused additional G-overhang elongation. This finding indicates that telomerase is able to extend the telomeric G-strand in $TEN1^{-/-}$ cells. Thus, TEN1 cannot be required for telomerase recruitment or G-strand extension.

**CTC1 and STN1 are sufficient to terminate telomerase action.** Given that CST inhibits telomerase activity[13], the above results imply that key phenotypic differences in the effect of CTC1 versus TEN1 disruption (i.e., excessive G- overhang elongation, damage signaling and growth arrest in $CTC^{-/-}$ cells) reflect a require-ment for CTC1 but not TEN1 to limit telomerase activity. If this is the case, chemical inhibition of telomerase should cause $CTC1^{-/-}$ cells to resemble $TEN1^{-/-}$ cells in terms of G-overhang length and telomere shortening due to the remaining deficit in C-strand fill-in. To test this prediction, we cultured CTC1 condi-tional cells with the telomerase inhibitor BIBR 1532 for 14 days prior to harvest for telomere length and G-overhang analysis (Supplementary Fig. 3c–e). The telomere length analysis revealed that the BIBR treatment indeed prevented telomere elongation in $CTC1^{-/-}$ cells and they instead exhibited gradual telomere shortening similar to $TEN1^{-/-}$ cells (Fig. 3c). G-overhang ana-lysis revealed that the extent of overhang elongation was also reduced to the level observed after TEN1 disruption (Fig. 3f–g). These results directly demonstrate that much of the G-strand growth in $CTC1^{-/-}$ cells results from telomerase action and thus a key role of CTC1 is to prevent excess telomerase-mediated repeat addition. Since the excess G-strand elongation is prevented in cells that lack TEN1 but retain CTC1 and STN1 expression, the results also indicate that either a CTC1-STN1 heterodimer or CTC1 alone is sufficient to restrain telomerase action and TEN1 is dispensable.

**CTC1 and STN1 are sufficient for telomere localization.** Both CST and telomerase interact with the shelterin subunit TPP1 and both bind ssDNA on the G-overhang[13,21,34,35]. Thus, an obvious way for CST to limit telomerase action would be through direct

competition for G-overhang binding and/or a mutually exclusive interaction with TPP1. Either mechanism would require CTC1-STN1 complexes or CTC1 alone to localize to telomeres in the absence of TEN1. We therefore used chromatin immunopreci-pitation (ChIP) to examine if this can occur.

For the ChIP, we generated TEN1 conditional cells that expressed FLAG-CTC1 or FLAG-STN1, and CTC1 conditional cells that expressed FLAG-STN1 or FLAG-TEN1 (Supplementary Fig. 4a). We then examined CTC1 or STN1 telomere localization in $TEN1^{-/-}$ cells and STN1 localization in $CTC1^{-/-}$ cells. CTC1, STN1, and TEN1 were precipitated from cross-linked chromatin with FLAG antibody and telomere association was assessed by hybridization with telomere probe (Fig. 4a, Supplementary Fig. 4c). The analysis revealed that loss of TEN1 did not decrease CTC1 or STN1 telomere association but rather caused a modest (not statistically significant) increase in association (Fig. 4b–c, Supplementary Table 2). This finding is consistent with previous studies indicating that CTC1 and STN1 each interact with TPP1[13]. In contrast, telomere association of STN1 and TEN1 was greatly diminished in the absence of CTC1 (Fig. 4d–e, Supple-mentary Fig. 4d). These results indicate that STN1-TEN1 localization depends on CTC1 and that CTC1 disruption causes loss of the entire CST complex from the telomere.

Given the importance of CTC1 for STN1 localization, we next asked whether CTC1 alone can associate with the telomere. To prevent CTC1 interaction with STN1 we generated CTC1 conditional cells expressing FLAG-CTC1 with a seven amino acid deletion (1196Δ7) previously shown to disrupt STN1 binding[36] (Supplementary Fig. 4b). We then asked if CTC11196Δ7 could localize to telomeres after disruption of endogenous $CTC1$ (Fig. 4f). ChIP analysis showed that the mutant exhibited much reduced telomere localization relative to WT CTC1 (Fig. 4g). This result implies that both CTC1 and STN1 are both necessary for CST to localize to telomeres.

**CTC1-STN1 retains ssDNA and ss-dsDNA binding activity.** Given that CTC1 and STN1 co-localize to telomeres, we were curious why CTC1-STN1 (CS) complexes are sufficient to limit telomerase action but insufficient for C-strand fill-in. A possible explanation is that CS binds ssDNA with much lower affinity than the full CST complex: e.g., μM versus nM binding[13,15,21,37]. The remaining interaction with TPP1 might be sufficient to terminate telomerase action, but inadequate to engage pol α for C-strand synthesis. To examine CS binding to DNA, we generated recom-binant protein and monitored binding by electrophoretic mobility shift assay (EMSA).

The EMSAs revealed that CS bound both telomeric and non-telomeric substrates with only modestly lower efficiency than the full CST complex (Fig. 5a). As previously reported, ST (STN1-TEN1) bound DNA only weakly[26]. To obtain a more quantitative comparison of CS and CST binding, we determined Kd(app) (Table 1 and Supplementary Fig. 5c–f). The Kds revealed a 5–13 fold reduction in CS binding affinity relative to CST, nonetheless, CS still exhibited high (nM) affinity binding. Interestingly, the relative change in CS affinity for an 18 nt substrate was less than the change for a 36 nt substrate. In the context of a CST complex, TEN1 and STN1 are positioned close to the DNA 3′ end and only CTC1 is expected to contact the 5′ region of a 36 nt substrate[21]. Thus, the magnified effect of TEN1 removal on binding to the 36 nt substrate, suggests that loss of TEN1 alters the architecture of the CS heterodimer and hence CST1 contacts with ssDNA.

We recently found that CST has a ss-dsDNA junction recognition activity that stabilizes binding to substrates of suboptimal length (e.g., 18 nt non-telomeric or 10 nt telomeric G-strand DNA)[20]. Since junction binding may be important for

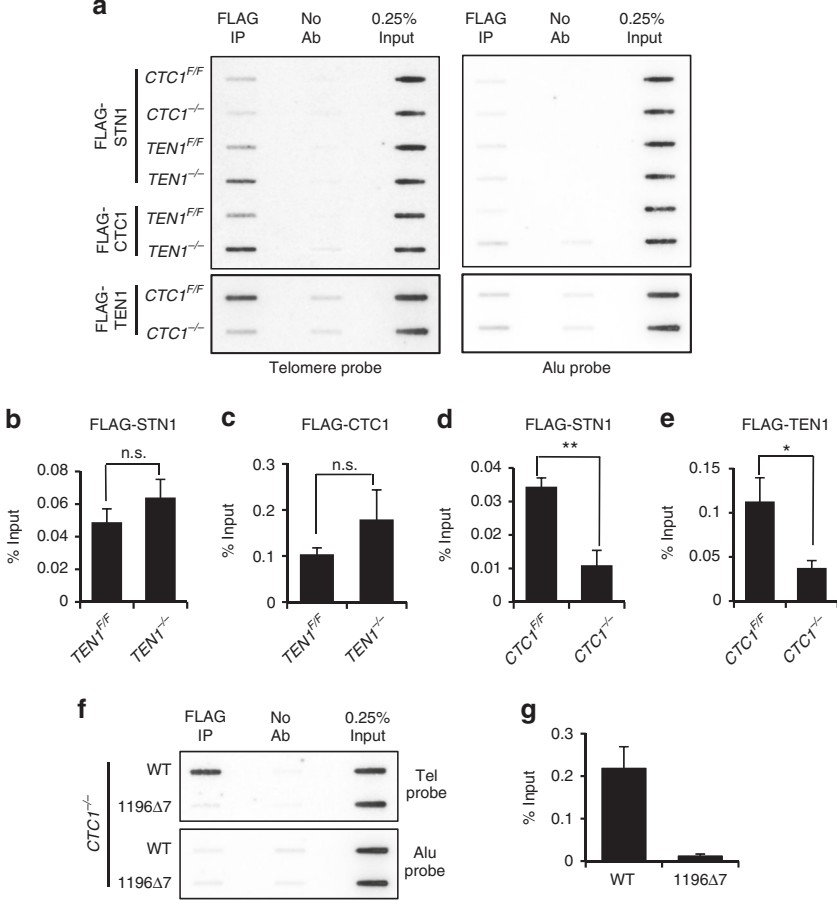

**Fig. 4** CTC1-STN1 localizes to telomeres in the absence of TEN1. **a–e** ChIP showing exogenous FLAG-STN1, FLAG-CTC1, or FLAG-TEN1 localization at telomeres after TEN1 or CTC1 removal. **a** Representative slot blots to quantify FLAG-STN1, FLAG-CTC1, or FLAG-TEN1 localization to telomeres. Cells were grown with/without tamoxifen for 7 days. **b–e** Quantification of ChIP data for FLAG-STN1 (**b**) or FLAG-CTC1 (**c**) in TEN1 conditional cells, FLAG-STN1 (**d**) or FLAG-TEN1 (**e**) in CTC1 conditional cells. Alu probe served as a negative control. **f–g** ChIP analysis showing the CTC1 1196Δ7 mutant localizes to telomeres. $CTC1^{-/-}$ cells expressed FLAG-CTC1 WT or FLAG-CTC1Δ7 and were grown with tamoxifen for 7 days. **f** Representative slot blots. **g** Quantification of ChIP data. $N = \geq 3$ independent experiments, Error bars indicate mean ± SEM. *$P < 0.05$; **$P < 0.01$; NS not significant

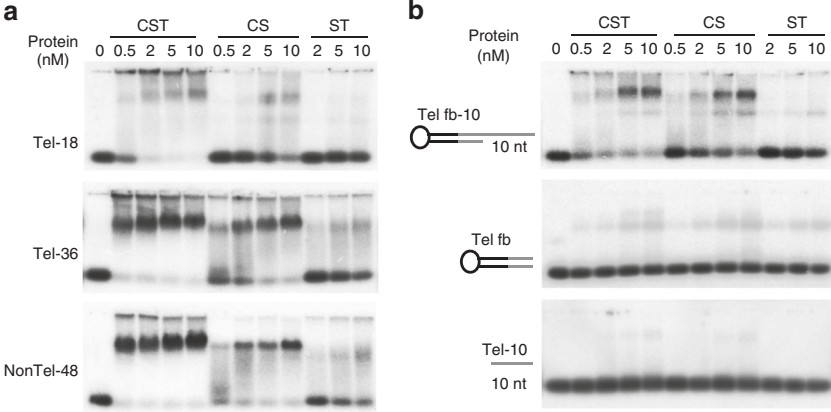

**Fig. 5** CTC1-STN1 binds ssDNA and ss-dsDNA junctions with reduced affinity. **a** EMSAs showing CST, CS, or ST binding to 18 or 36 nt telomeric G-strand (Tel-18, Tel-36) or 48 nt non-telomeric (NonTel-48) ssDNA substrates. Reactions contained 0.1 nM DNA and the indicated concentrations of protein. **b** EMSAs showing CST, CS, or ST binding to ss-dsDNA telomeric junction substrate with 10 nt 3′ overhang, same substrate lacking the overhang or 10 nt ssDNA of same sequence as the overhang. Black line: non-telomeric, gray: telomeric sequence; fb fold-back

CST to position pol α to initiate C-strand synthesis, we asked whether CS complexes also recognize ss-dsDNA junctions. EMSAs were used to assess binding to two substrates: one had 10 nt telomeric G-strand overhang, the other had an 18 nt mixed sequence overhang. Both substrates were formed from a single

(fold-back) oligonucleotide which self-hybridized to give 15 bp dsDNA and a 10 or 18 nt 3′ overhang[20]. The EMSAs indicated that CS bound both substrates with only modestly (5-fold) decreased efficiency relative to the full CST complex (Fig. 5b, Table 1, Supplementary Fig. 5b). As expected, affinity for the

| Table 1 Kd(app) for CST or CS binding to the indicated substrates | | | |
|---|---|---|---|
| Substrate | Kd(ap) (nM) | Kd(ap) (nM) | Fold change |
| | CST | CS | |
| Tel-18 | 0.21 ± 0.07 | 1.04 ± 0.46 | 5.0 |
| Tel-36 | 0.02 ± 0.03 | 0.33 ± 0.08 | 13.3 |
| Non Tel-48 | 0.01 ± 0.005 | 0.08 ± 0.05 | 7.1 |
| Tel fb-10 | 0.08 ± 0.35 | 3.76.21 ± 2.71 | 4.7 |
| N = ≥3 independent experiments | | | |

corresponding 10 nt telomeric or 18 nt non-telomeric fully ssDNA was very low[20]. These results indicate that CS binding to a suboptimal ssDNA substrate is also stabilized by an adjacent region of dsDNA. Thus, TEN1 is not the determinant for junction recognition.

**TEN1 determines DNA binding stability and dynamics.** Since CST harbors multiple DNA-binding sites (OB folds) that may individually bind and release DNA, much of CST function could be determined by the dynamic nature of CST binding rather than the macroscopic binding affinity of the whole complex[20]. We therefore turned to single-molecule FRET (smFRET) to examine CS binding as this technique provides an excellent way to visualize individual binding events and dynamics. We used the same partial duplex substrate used to previously study CST binding[20]. It was prepared by annealing a 3′ Cy3-labeled 36 nt oligonucleotide to an 18 nt 5′ Cy5-labeled oligonucleotide (Fig. 6a). The DNA was then anchored to NeutrAvidin-coated slides through a 3′ terminal biotin. Although this study focuses on CS binding at telomeres, we used a non-telomeric substrate because it reports simultaneously on junction binding and binding to ssDNA. A 10 nt overhang is needed to detect junction binding with a telomeric substrate (Fig. 5b) but this is too short to detect loss of FRET.

FRET was measured using a prism-type total-internal-reflection microscope and data were plotted to give FRET histograms showing Gausian probability distribution. In the absence of protein, a high FRET peak predominated due to the flexibility of the 18 nt ssDNA bringing the Cy3 (donor) and Cy5 (acceptor) labels into close proximity (Fig. 6b). When CST was added to the slide we saw the expected FRET efficiency switch from ~0.7 to ~0.15, indicating a time-averaged increase in the distance between the Cy3 and Cy5 labels due to protein binding[20] (Fig. 6b, Supplementary Fig. 6a). Interestingly, the effect of CS addition depended on the timing of data acquisition. If CS was added to the slide for 10 min and then removed by flushing the slide with imaging buffer prior to data acquisition, we observed only the high FRET peak, indicating a lack of protein binding (Supplementary Fig. 6b). In contrast, when imaging was initiated immediately after CS addition we detected a low FRET (E ~0.15) peak (Fig. 6b). However, the fraction of substrate molecules exhibiting the switch in FRET efficiency was much lower than after CST addition. These results suggested that CS binding was quite unstable because the protein appeared to dissociate from the substrate when excess protein was flushed from the slide.

To examine CS binding more closely, we performed a real-time analysis to monitor the change in FRET signals from individual DNA molecules with time after protein addition (Fig. 6c and Supplementary Fig. 6c–g). Imaging was initiated immediately after protein addition. Comparison of the traces obtained with CS and CST revealed interesting similarities and differences. For both complexes, binding resulted in a sharp one-step transition to the low FRET state. The sudden change in FRET supports rapid

binding of multiple individual binding sites instead of slower step-by-step association[38]. Likewise, examination of the individual Cy3 and Cy5 signals revealed anti-correlated fluorescence after binding of either complex with the Cy3 donor emission increasing as the Cy5 acceptor emission decreased. The increase in Cy3 fluorescence indicated that the Cy3-labeled DNA remained associated with the slide-anchored Cy5-labeled DNA. While we cannot rule out partial melting of the DNA duplex, this result indicates that CS, like CST, is unable to completely separate the two DNA molecules as full strand-melting would have led to loss of the Cy3 signal from the evanescent field (Fig. 6a). Thus, the smFRET confirms that CS retains junction recognition activity as even with partial melting of the dsDNA, the length of ssDNA would remain too short for CS to bind (Supplementary Fig. 5b)[20].

The most striking difference in the CS and CST real time traces lay in the stability of binding. As observed previously for CST, many of the traces showed protein dissociation and rebinding during the recording. However, CS dissociated more frequently than CST (~59% of CST and 67% of CS traces exhibited dissociation events) and the dwell time (δ) for individual binding events was 43% lower for CS (Fig. 6d, Supplementary Fig. 7a–b). Traces obtained with CS also showed a >2-fold increase in the appearance of partial and very transient troughs in the FRET signal (<4 s duration) where the signal decreased to 0.4–0.5 instead of to 0.15 as observed during the longer lived CS or CST binding events (Fig. 6c, e, Supplementary Fig. 7c, Supplementary Table 3). We suspect that these transient troughs in the FRET signal reflect partial binding events which failed to progress to a full interaction. Overall, the smFRET data indicate that TEN1 plays an important role in stabilizing CS binding to ssDNA. This binding stabilization could occur in several ways. Since TEN1 cross-links to ssDNA[21], it may provide CST with an additional DNA-binding domain. TEN1 may also enhance CS binding by altering the architecture of the CS complex in a manner similar RPA3 which acts as a hub for RPA complex assembly[39]. Given the importance of TEN1 for ssDNA binding stability, it is noteworthy that loss of TEN1 has no significant effect on CS telomere localization (Fig. 4b). This finding suggests that CST telomere localization may depend on CTC1 and STN1 interaction with the shelterin subunit TPP1.

## Discussion

We have used TEN1 and CTC1 knockout cell lines to delineate the roles of individual CST subunits in telomere maintenance (Fig. 6f). We show that CTC1-STN1 (CS) terminates DNA synthesis by telomerase to prevent overextension of the telomeric G-strand. Both subunits are required for this activity whereas TEN1 is dispensable. The regulation of telomerase by CS is essential for cell proliferation because in the absence of CS, overextension of the G-overhang causes telomeric DNA damage signaling due to POT1 exhaustion and RPA binding[4,40]. The role of TEN1 is to stabilize CS binding to ssDNA and to enable C-strand fill-in following G-strand extension by telomerase. The contribution of TEN1 is essential for telomere length maintenance as disruption of C-strand fill-in by TEN1 removal results in progressive telomere shortening similar to that caused by telomerase deficiency.

It is striking that CS is fully capable of limiting telomerase action despite loss of C-strand fill-in because the physical act of C-strand synthesis appears to contribute to telomerase inhibition in ciliates and yeast[41–44]. Clearly this is not the case in humans. It is also notable that human TEN1 is unnecessary for telomerase regulation given that TEN1 or the STN1-TEN1 complex is responsible for terminating telomerase activity in *Arabidopsis* and

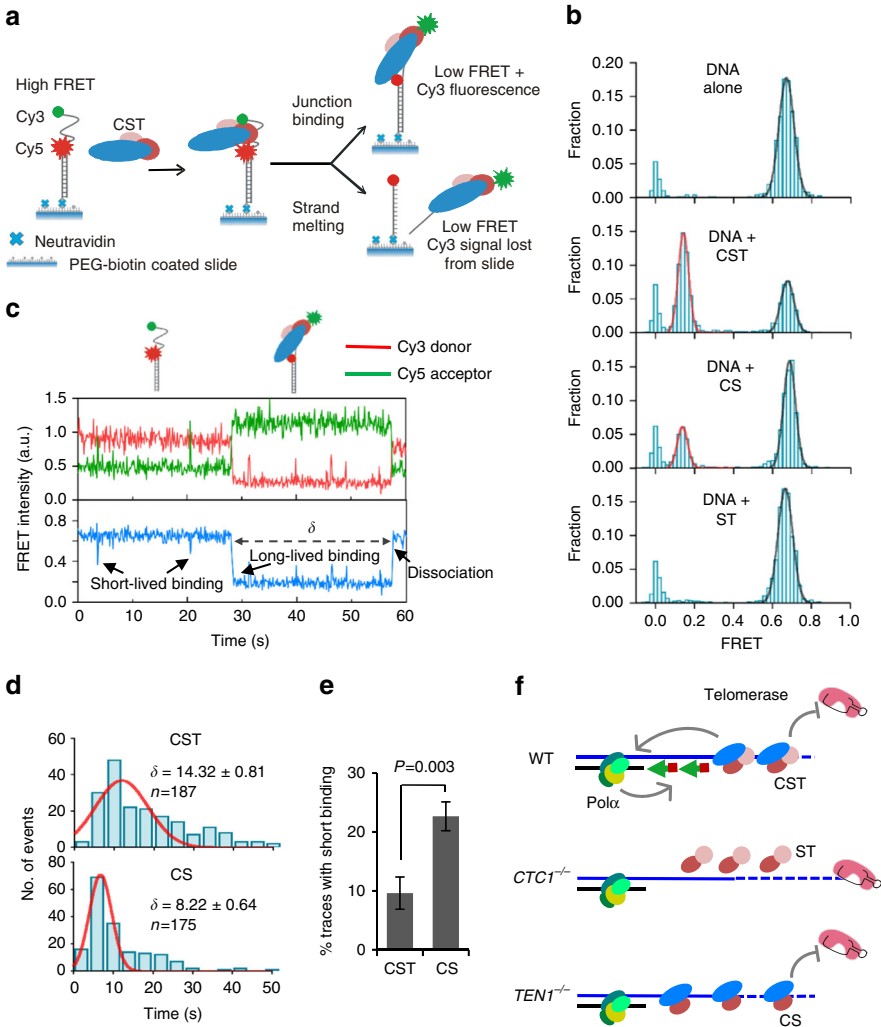

**Fig. 6** Removal of TEN1 destabilizes CTC1-STN1 binding. Single molecule FRET analysis of CST, CS or ST binding to DNA junction substrate with 18 nt overhang. **a** Cartoon showing design of the non-telomeric junction substrate and anticipated FRET signals in the presence or absence of CST. In the absence of CST the flexibility of the 18 nt ssDNA will bring the Cy3 donor and Cy5 acceptor into close proximity and give a high FRET signal. If CST binds without melting the anchoring DNA duplex, the high FRET signal will be lost but emission from the Cy3 donor (green) will be retained. If CST melts the DNA duplex, both the high FRET and the Cy3 donor signal will be lost. **b** FRET histograms generated from FRET measurements of >4000 single molecules. The zero FRET peak reflects DNA molecules with an inactive or missing Cy3 label. **c** smFRET real-time trace showing change in individual Cy3 and Cy5 signals (top) and FRET (bottom) with time. Dwell time ($\delta$), long-lived binding, short-lived binding and dissociation events are indicated. **d** Distribution of dwell times for CST or CS binding before protein dissociation. Histogram shows number of binding and dissociation events within the indicated 4 s time intervals. Red line shows the calculated Gaussian distribution. **e** Percent of traces showing a short-lived trough in FRET signal after CS or CST addition. Error bars indicate SEM. **f** A model depicting effects of CTC1 or TEN1 loss on ability of remaining CST subunits to restrain telomerase action and mediate C-strand fill-in

yeast[45–48]. It appears that while some form of (C)ST complex is used to coordinate telomerase and pol α activity at telomeres in many species, the precise roles of the individual subunits has evolved quite rapidly. In vitro studies with human CST point to several mechanisms for telomerase regulation by CST including sequestration of the ssDNA substrate and preventing telomerase stimulation by POT1-TPP1[13]. Given that CS and CST both prevent G-strand overextension despite the reduced stability of CS binding to ssDNA, our results suggest that CS/CST interaction with TPP1 is the key event needed to terminate telomerase. This interaction might lead to telomerase displacement due to direct competition for TPP1 binding[10,11]. Alternatively, CST binding to TPP1 might alter a post-translational modification necessary for telomerase–TPP1 interaction[49].

During C-strand fill-in, the role of CST is to enable DNA synthesis by pol α. Since pol α localizes to the telomere even after STN1 depletion, CST does not appear to play a role in pol α recruitment[31]. Instead, the complex stimulates primer synthesis, most likely at multiple points in the reaction[14,16,50]. One role of CST is to enhance the switch between RNA priming and DNA synthesis through a direct interaction between STN1 and the p70 subunit of pol α[16]. It is intriguing that TEN1 is absolutely required for C-strand fill-in even though only CTC1 and STN1 appear to interact directly with pol α (Supplementary Fig. 8)[21] and it is STN1 that directs the RNA to DNA switch[16]. Our recent discovery that CST binds DNA dynamically[20] leads us to suggest that destabilization of CS binding and the resulting changes in binding dynamics after TEN1 removal renders CS unable to

properly engage pol α on the G-overhang for the primer synthesis reaction.

Like RPA, CST can melt G4 DNA structure and mediate protein displacement from ssDNA[20]. In the case of RPA, these properties stem from the dynamic dissociation and reassociation of individual OB folds from the ssDNA[25]. Mutations that alter the binding of individual OB-folds can selectively affect certain aspects of RPA function most likely by altering its capacity to melt DNA structure or direct loading–unloading of partner proteins[23,24,51]. In the case of CST, altered binding dynamics due to TEN1 removal could disrupt C-strand synthesis at multiple levels. Decreased melting of G4 DNA might impede pol α loading on the overhang, altered interaction of CS with the overhang might leave STN1 incorrectly positioned to enhance the switch from RNA to DNA synthesis, or the interaction with pol α might cause CS to dissociate from the overhang without stimulating DNA synthesis. The latter possibility seems particularly likely given that STN1 contacts pol α through its OB fold[16]. As the STN1 OB fold also stabilizes CST binding to ssDNA[21], the interaction with pol α could weaken CS binding to the G-overhang and hence make TEN1 more important for binding stability during pol α activation.

While our analysis of CST function utilized cancer cells, other studies indicate that CST is also required for telomere maintenance in human primary cells[31]. As in cancer cells, the role in C-strand fill-in is important as CST is needed to prevent G-overhang elongation and increased telomere shortening due to nuclease resection of the telomeric C-strand[31]. Given the importance of CST for telomere maintenance, it is not surprising that mutations in CST subunits cause human disease. Mutations in CTC1 and STN1 cause the telomeropathy and genome instability syndrome Coats plus[52,53]. The mutations are hypomorphic and patients retain residual CST function. Thus far, patient mutations have not been found in TEN1. One has to wonder whether this reflects the small size and essential nature of TEN1, perhaps mutations that disrupt TEN1 association with either STN1 or the telomeric DNA cannot be tolerated due to the essential nature of the C-strand fill-in reaction.

## Methods

**Cell culture and generation of *TEN1^{F/F}* cells.** HEK293T cells were grown in DMEM and HCT116 cells in McCoy's medium supplemented with 10% FBS, antibiotics, and glutamine. Both cell lines were originally obtained from the ATTC and were tested for mycoplasma contamination. The HCT116 cells used to make the CTC1 and TEN1 conditional cells were $p53^{+/+}$ Supplementary Fig. 1c–d). CTC1 conditional cells were generated using adeno-associated virus to introduce LoxP sites into introns 4 and 5 and stable integration of Cre-ER[4] Gene disruption was achieved by growth in tamoxifen. Cells with conditional TEN1 disruption were generated using CRISPR/Cas9 genome editing to introduce LoxP sites into introns two and three of the *TEN1* gene locus. Each LoxP site was introduced by a separate round of genome editing. A single clone from the first round of genome editing was used for the second round of editing. HCT116 cells were transiently transfected with PX458M gRNA/Cas9 vector which included a GFP expression element (provided by CCHMC Transgenic Animal and Genome Editing Core Facility) and a donor oligonucleotide containing the LoxP site, ~60 nt of sequence homologous to the relevant intron of the *TEN1* gene locus, and an Sph1 restriction digestion site (sequence given below). The sequences of the gRNAs were GAGCAGCCTGGA AGGTCA and GTGAAGAGACAATCCCCCAT. Fortyeight hour after transfection, GFP positive cells were sorted into 96 well plates by flow cytometry to isolate single clones. To screen for correct genome editing, DNA was isolated from each clone and analyzed by PCR using primers 5′-CTGAAATGTCTCAAGTAAACA GCAG and 5′-ATGAGCCACCACACCTGATC for the intron two insertion, and 5′-ACTCAAAGACAGGGTGGCTG and 5′-GATGGAGGTTGCAGTGAGCT for the intron three insertion. The PCR product was digested with Sph1 and positive clones identified based on the correct size and digestion pattern. To introduce Cre recombinase, retrovirus was generated by co-transfecting 293T cells with CreERT2-puro (Addgene, 22776), gag-pol, and env. Viral supernatant was used to infect a single clone of HCT116 *TEN1^{F/F}* cells. Individual cells were isolated by cell flow cytometry and selected with puromycin. Multiple clones obtained after the Cre-ER integration were isolated and screened for TEN1 gene disruption and telomere phenotypes (Fig. 1, Supplementary Fig. 2). For gene disruption, tamoxifen (Sigma,

H7904) was added to 10 nM to induce Cre activity. TEN1 disruption was verified by PCR with primers 5′-CTGAAATGTCTCAAGTAAACAGCAG (located in intron 2) and 5′-GATGGAGGTTGCAGTGAGCT (located in intron 3) and by Western blot analysis.

To guard against off-target effects during gene targeting, we performed Sanger sequencing on the *TEN1* locus and on five additional loci that showed the next highest homology to the individual gRNAs. Sequencing of the *TEN1* gene locus verified correct insertion of each LoxP site while sequencing of the five other loci was to ensure there was no incorrect LoxP insertion or other genome editing at these sites. We also generated a rescue cell line where an exogenous TEN1 cDNA was introduced into the TEN1 conditional cells. This cell line was used to verify wild-type phenotypes after endogenous TEN1 disruption. To overexpress TEN1, STN1 or hTERT, *TEN1^{F/F}* cells were transfected with pMIT vector encoding TEN1/ FLAG-STN1 or pBabe vector encoding hTERT and cells were selected by flow cytometry for Thy1 expression for pMIT vector or hygromycin for pBabe vector. See Supplementary Table 4 for full list of primers and oligonucleotides used to generate and verify cell lines.

**Growth curves.** *CTC1^{cond.}* and *TEN1^{cond.}* cells grown in media with or without TAM. A total of $1 \times 10^5$ cells were plated in triplicate and grow for 3 or 4 days. Cells were counted using trypan blue exclusion, and $1 \times 10^5$ cells were reseeded for the next time point. Total cell number was extrapolated using a value that was the cell count/fraction of cells used to reseed previous time point. The experiment was performed twice.

**Telomere length and G-overhang analysis.** Genomic DNA was digested overnight with HinfI, MspI, and RsaI, then separated in agarose gels. The gels were dried, denatured and hybridized with $^{32}$P-labeled $(TA_2C_3)_3$ probe. For G-overhang analysis, control samples were treated with ExoI for 48 h prior to restriction digestion, and the gels were initially hybridized with $(TA_2G_3)_3$ probe under nondenaturing conditions. For telomere length analysis, TRF signal intensity was quantified by PhosphorImager, and mean telomere length was determined by dividing each lane into boxes using ImageQuant and applying the formula $\Sigma$ *Sig* /$\Sigma$ (SigI/LI), where Sig is the sum of the signal from all 100 boxes, SigI is the signal in an individual box, and LI corresponds to the average length of the DNA in that box as determined using DNA markers and a standard curve where distance migrated was plotted against DNA length[54].

**Telomere FISH and γ-H2AX staining.** FISH was performed on MeOH/acetic acid fixed metaphase. Samples were hybridized with CENPB-Cy3 pan-centromere PNA probe (5′ATTCGTTGGAAACGGGA, Biosynthesis) in addition to TelC-Alexa488 PNA G-strand probe (5′-CCCTAACCCTAACCCTAA, Biosynthesis) or TelG-Cy3 PNA C-strand probe (5′-GGGTTAGGGTTAGGGTTA, Biosynthesis). Images were taken at a constant exposure time. For quantitative measurement of telomere length (qFISH), telomere fluorescence intensity was integrated using the TFL-TELO program. Signal free ends (SFE) and telomere fusions are quantified by eye.

For γ-H2AX and telomere staining, cells were synchronized with colchicine (0.1 ng/μl) for 2 h and then collected for staining. The cells were first swelled with hypotonic buffer (0.2% KCl and 0.2% tri-sodium citrate) and spread on slides by cytospin, Cells were then fixed with 4% formaldehyde in PBS and permeablized in KCM buffer (120 mM KCl, 20 mM NaCl, 10 mM Tris pH 7.5, 0.1% Triton X-100). Slides were blocked in antibody-dilution buffer (20 mM Tris pH 7.5, 2% BSA, 0.2% fish gelatin (G7041, Sigma), 150 mM NaCl, 0.1% Triton X-100, 100 μg/ml RNase A (Sigma R6513)) at 37 °C for 15 min. For γH2AX staining, slides were incubated with 1:500 dilution of γH2AX antibody (Millipore, 05-636) overnight at 4 °C in a humidified chamber and then with 1:1000 dilution of goat anti-mouse secondary antibody (Invitrogen A21422). FISH was performed TelC-Alexa488 PNA G-strand probe[55]. γH2AX foci were scored on ~100 chromosomes for each experiment.

**TRAP assay.** TRAP assays were performed using Millipore TRAPeze Telomerase Detection Kit. Briefly, cells were lysed with $1 \times$ CHAPS lysis buffer with RNase inhibitor. Protein concentration was quantified using Pierce BCA protein assay kit. Reactions contained 50–400 ng protein extract TRAP buffer, dNTP mix, TS primer, TRAP primer mix, and Taq polymerase. Telomerase extension was performed at 30 °C for 30 min, followed by PCR amplification. PCR products were detected by electrophoresis and SYSB green staining. ImageQ software was used to quantify the signal in each lane. For quantification, the total telomere repeat signal from each lane was divided by the internal control signal from the same lane. The normalized signal from the *TEN1^{F/F}* or *CTC1^{F/F}* cells was then set to 1.

**Chromatin immunoprecipitation.** Cells were fixed with 1% formaldehyde for 15 min, treated with 200 mM glycine, pelleted by centrifugation, suspended in swelling buffer (25 mM HEPES, pH 7.9, 10 mM KCl, 1.5 mM MgCl₂, 1 mM EDTA, 1 mM DTT, 0.25% Triton X-100 and home-made protease inhibitor cocktail containing 1 μg/ml pepstatin, 5 μg/ml leupeptin, 1 μg/ml E64, 2 μg/ml aprotinin, and 5 μg/ml antipain, all from SIGMA) and incubated for 10 min on ice, resuspended in sonication buffer (50 mM HEPES, pH 7.9, 150 mM NaCl, 1 mM EDTA, 0.1% Sodium deoxycholate, 0.1% SDS, 1% Triton X-100, and protease inhibitors) and sonicated for 20 min. For immunoprecipitation, samples containing

supernatant (0.3 mg protein), antibody (3 μg FLAG M2, Sigma A2220) and 20 μg bacterial DNA were incubated overnight at 4 °C. Protein A/G PLUS agarose beads (Santa Cruz sc-2003) were added to samples for 1 h, the beads were washed and eluted in 450 μl elution buffer (1% SDS, 0.1 M NaHCO3). Cross-linking was reversed by incubation at 65 °C overnight. The eluate was brought to 10 mM EDTA, 40 mM Tris-HCL, pH 6.8, and treated with 5 μg RNase A then 40 μg protease K, and the DNA purified by phenol-chloroform extraction. DNA was also purified from a sample of each input chromatin. The samples were analyzed by slot-blot hybridization with[32] P-labeled (TA$_2$C$_3$)$_3$ probe and the signal determined by Phosphorimager. To quantify the amount of DNA precipitated with each antibody, the background signal from the no antibody control was subtracted from that of the experimental samples and the amount of precipitated DNA was calculated as a percentage of the corresponding input DNA.

**Western blot analysis**. Cells were lysed in NP-40 lysis buffer (20 mM Tris pH 8.0, 100 mM NaCl, 1 mM MgCl$_2$, and 0.1% Igepal) and samples were separated by SDS-PAGE and transferred to nitrocellulose. The membrane was blocked with 5% milk in TBS-Tween (50 mM Tris pH 7.5, 150 mM NaCl, 0.1% Tween 20) and incubated with antibody to 3xFLAG (M2 Sigma A2220, 1:1000 dilution), actinin (Santa Cruz, 1:10000 dilution), CTC1 (Millipore MABE1103, clone C482, 1:500 dilution). The TEN1 and STN1 antibodies were home-made. Validation is described in Kasbek et al.[30]. Uncropped blots are shown in Supplementary Fig. 9.

**Protein purification**. HEK293T cells were co-transfected with pcDNA3 expression vectors encoding FLAG-CTC1, FLAG-STN1, and TEN1 using polyethylamine (PEI). Cells were harvested 72 h later, and lysed with NP-40 lysis buffer (0.1% Igepal, 20 mM Tris-HCl pH 8.0, 100 mM NaCl, 1 mM MgCl$_2$, and home-made protease inhibitor cocktail). The supernatant was incubated with FLAG-M2 beads (Sigma A2220) for 2 h. The beads were then washed and eluted with 3 × FLAG peptide (Sigma F4799). The protein concentration was quantified by silver staining using BSA as a standard.

**Electrophoretic gel mobility shift assays**. Indicated amounts of protein complex were incubated with 0.1 nM $^{32}$P-labeled oligonucleotide (Supplementary Table 5) in 25 mM Tris-HCl pH 8.0, 100 mM NaCl, 1 mM DTT for 30 minutes at RT. For KD(app) analysis, protein was incubated with 0.01 nM $^{32}$P-labeled oligonucleotide in 25 mM Tris-HCl pH 8.0, 100 mM NaCl, 1 mM DTT for 18 h at 4 °C. Samples were separated in 0.7% agarose gels with 1× TAE and quantified by Phosphorimager. Small boxes were created to encompass the free DNA, larger boxes were created to encompass the portion of the gel containing DNA-protein complexes (the sizes were equal across lanes). For each box, the background signal from an equivalent area of a blank lane was subtracted. The signal from the bound DNA was then divided by the signal from the free DNA to get the percent bound. Replicates were performed with different protein preparations.

**Single-molecule fluorescence resonance energy transfer**. Slides coated with a mixture of 97% mPEG and 3% biotin PEG, flow chambers were then assembled using strips of double-sided tape and epoxy. 30 μl of 0.2 mg/ml NeutrAvidin was flowed into each empty flow chamber and incubated for 5 min. The excess NeutrAvidin was then washed out and partial duplex DNA molecules (Supplementary Table 5) flowed into the slide and immobilized by biotin-neutravidin interaction. Excess oligonucleotide was then washed out with T50 buffer (10 mM Tris-HCl pH 8.0, 50 mM NaCl). To detect binding, 2 nM protein complex in imaging buffer (25 mM Tris-HCl pH 8.0, 1 mM DTT, 150 mM NaCl, 0.8 mg/ml glucose oxidase, 0.625% glucose, 3 mM Trolox and 0.03 mg/ml catalase) was added to the slide and imaging was initiated immediately. The whole procedure was carried out at room temperature. All smFRET experiments were performed with at least three separate protein preparations.

Data acquisition was performed using a prism-type total internal reflection fluorescence (TIRF) microscope equipped with a dual-laser excitation system (532 and 640 nm Crystal Laser) to excite Cy3 and Cy5. Fluorescence signals were collected by a water immersion objective lens and then passed through a notch filter to block out excitation beams. The emission signals from Cy3 and Cy5 were separated by a dichroic mirror and detected by the electron-multiplying charge-coupled device camera. For FRET histograms short (2 s) movies were recorded from 20 to 30 random locations. For real-time measurements, long movies (60 or 90 s) were recorded from 5 to 10 random locations. Data analysis was carried out by the smCamera software written in C++ (Microsoft) with FRET efficiency E, calculated as the intensity of the acceptor channel divided by the sum of the donor and acceptor intensities. FRET histograms were generated by using measurements from >4000 molecules and were fitted to Gaussian distributions with an unrestrained peak center position in Prism 7 (GraphPad Software).

**Code availability**. The custom software can be downloaded from https://cplc.illinois.edu/software/.

**Statistical methods**. Data were analyzed by two-tailed Student's $t$ test. $P$-values: *$P < 0.05$, **$P < 0.01$.

**Data availability**. All data generated or analyzed during this study are included in this published article and its supplementary information files and can be provided by the authors upon suitable request.

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

## Acknowledgements

We would like to thank Shiva Senthil Kumar for help with TRAP assays, Birgit Ehmer for assistance with cell sorting, Yueh-Chiang Hu and the CCHMC transgenic Animal and Genome Editing Core Facility for design of the CRISPR guide RNAs. This work was supported by National Institute of Health grant RO1 GM041803 to C.M.P. and start-up funds from the University of Cincinnati to J.D., S.J.H. was supported by T32 CA117846.

## Author contributions

Author contributions are as follows: X.F. designed project, performed experiments, and wrote sections of the manuscript. A.B., S.J.H. and Y.W. performed experiments. J.D. guided smFRET experiments. C.M.P. devised the project, guided experimental design, oversaw the research, and wrote sections of the manuscript.

## Additional information

**Competing interests:** The authors declare no competing interests.

