## [Peer Review File · Nature Communications]

Reviewers' comments:

Reviewer #1 (Remarks to the Author):

In this manuscript, authors attempt to dissect the contributions individual human CST subunits play in telomere length maintenance and G-overhang regulation. Using CRISPR/CAS9, they introduced loxP sites to CTC1 or TEN1 loci in HCT116 cells to permit conditional knockout of these genes. As previously documented in several of their previous papers, including one published in NAR (2017), deletion of CTC1 results in slight telomere damage, eliciting a DNA damage response that compromises cell growth. In addition, elongation of the 3' G-overhang while C-strand length decreases are also observed in this cell type. The novelty of the current submission is that TEN1^{-/-} cells do not exhibit telomeric DNA damage and cell growth arrest. Removal of TEN1 only leads to a slight increase in G-overhangs and telomere shortening due to failure to properly synthesize the C-strand. The authors conclude that TEN1 is required for C-strand synthesis. Using a series of biochemical and biophysical experiments, they suggest that both CTC1-STN1 are required for telomerase to terminate DNA synthesis on the G-strand while STN1-TEN1 functions to promote C-strand fill-in by DNA polymerase. Unfortunately, the TIN1 data is not very convincing, reducing my enthusiasm for this paper.

1. I always worry when important conclusions in this paper are drawn from only one CRISPR/CAS9 cell line, given the well documented off-target effects that can occur using this method of gene targeting. The authors claim they generated multiple clones but only chose one for analyses-why? They have to show telomere length data from at least one additional Tam-treated CTC1F/F and TEN1F/F cell line, as well as control CTC1F/F and TEN1F/F cell lines not treated with Tam. These controls are especially important since in Figure 3B and 3C, it appears that telomere lengths in CTC1^{-/-} and TEN1^{-/-} cells, even at day 0 of Tam treatment, are already significantly different from each other. Why? Additional controls as indicated above are needed to resolve this issue.

2. The authors show that telomeres lacking CTC1 activate a damage response and recruits γ H2AX. What is the p53 status in these cells? One would assume it has to be p53^{+/+} to elicit the growth arrest seen in the CTC1 null cells, but the authors need to show this data.

3. The novelty of this paper is the authors' suggestion that TEN1 regulates C-strand filling by DNA pol alpha. However, the data presented are rather unconvincing. I do not see any telomere overhang length change in TEN1 null cells after double thymidine block (Supp Figures 2C). The authors need to inhibit DNA pol a in this experiment to make these effects more apparent. Also, since TEN1^{-/-} cells have no growth issues, why not monitor them longer over many more passages to examine the status of the C-strand and the G-overhang? A TEN1F/F control cell line is needed in all of these experiments.

4. In Figure 3C, addition of the telomerase inhibitor BIBR reduces telomere length in CTC1 null cells, likely by inhibiting telomerase. However, as shown in Supp. Fig. 3C, addition of BIBR did not fully inhibit telomerase activity in these cells. If telomerase activity is still present in BIBR treated CTC1 null cells, how did these telomeres shorten?

5. Does TEN1 play a direct role in recruiting DNA pol alpha to the C-strand? Supp figure 6F suggest this, but it is necessary to show Pol a recruitment by CHIP in both TENF/F and TEN^{-/-} cells.

6. There's a disconnect between Figure 4 and 5/6. Figure 4 shows that STN1 and CTC1 localizes to telomeres robustly in vivo in the absence of TEN1. However, Figure 5/6 show that CS binding to in vitro telomeric substrates in the absence of TEN1 is unstable. So which is it-they can't be both correct?

Additional notes:

1. In Supp. Fig 1D, signal free end info need info for STN1F/F and CTC1F/F cells too.

Reviewer #2 (Remarks to the Author):

The human CST complex (CTC1-STN1-TEN1) is an important regulator of telomere homeostasis, with implicated roles in both regulation of telomere extension by the ribonucleoprotein reverse transcriptase telomerase as well as in orchestrating recruitment of the necessary replication proteins required for promoting C-strand fill-in following telomerase action. Although CST has been shown to be critical for telomere regulation in a variety of model systems, the precise division of labor between the components of the CST complex in mediating its various cellular activities has not been firmly established. Feng et al. present a detailed study of the human CST complex, using an impressive combination of cell-based assays, together with biochemical and biophysical analyses. The major conclusions of the work are that TEN1 is not required to limit telomerase activity, but is essential for C-strand fill-in synthesis. In the absence of TEN1, HCT116 cells retain the ability to continue to divide and do not elicit a DNA damage response as measured by gammaH2AX foci (unlike CTC1 negative cells), but do cause a comparable increase in the number of signal free ends and chromosome fusions as CTC1 negative cells. The distinct phenotypes of the TEN1 negative and CTC1 negative cell lines suggested these components of the CST complex serve non-overlapping roles and prompted further investigation by the authors. Using the same cell lines, the authors measured the change in G-strand overhang signal and find that removal of CTC1 has a significantly larger positive effect on the length of the G-strand overhang than that observed for the TEN1 negative cells. This result leads to the conclusion that TEN1 is not required to limit the action of telomerase at telomeres. The authors next turn to telomere length analysis and find that TEN1 removal does cause detectable telomere shortening, but not a decrease in telomerase activity, suggesting the effect of TEN1 removal is to limit C-strand fill-in. The finding that the CS complex was sufficient to limit telomerase activity prompted subsequent experiments using ChIP to test whether both CTC1 and STN1 were required for telomere localization. The results of the ChIP experiments support the notion that both CTC1 and STN1 are required for wild type levels of telomere localization. Having investigated the properties of CST in their cell lines, the authors take the commendable step to characterize the DNA binding properties of CST and its sub-complexes in vitro. The authors first turn to EMSA experiments to test the binding of CST, CS, and ST to various model telomere DNA substrates. The results of the EMSA analysis suggest that the presence of TEN1 stabilizes the protein-DNA complex as compared with the CS complex alone. Finally, the authors present yet another complementary experiment, using single-molecule FRET to analyze the binding dynamics of the various recombinant CST, CS and ST complexes. The authors observe smFRET signals that they interpret to represent short-lived and long-lived protein bound states, with the presence of TEN1 supporting more long-lived binding events. Overall, the manuscript is clearly written, the experiments well designed, and with a few exceptions (see below) the conclusions are largely supported by the experimental data. Therefore, if the authors can address the specific points below in a revised manuscript, I expect this study will be appropriate for publication in Nature Communications.

Major Points:

1. There are several areas where a description of certain methods should be better explained in the paper. This is true for the TRAP assay results shown in Fig. 3A - how was the assay done and what is being quantified in this bar plot? Also, with respect to the purification of the CST complex, there appears to be no description of how these complexes were made in the methods section. The only details appear to be included in the figure legend of Fig. S5 where it is stated that the proteins were coexpressed in 293T cells and purified by FLAG pull down. More details for the production of the recombinant complexes should be included given how many experiments are done with them. On this same topic, do the authors have any estimates of the stoichiometry of their complexes? The TEN1 protein appears to be limiting in their gel, but is this just a consequence of weak staining due to its small size. Better characterization and description of the protein complexes would be appreciated and improve the paper.

2. In Fig. 3C, the authors should consider showing the same experiment with CTC1 F/F cells for proper comparison, since BIBR treatment should result in telomere shortening regardless of CTC1 knockdown. Also, why is the 14-day time point excluded? It seems that day 10 is where the G-strand elongation is starting to become noticeable in the -BIBR experiment, and likewise, at day 10 is when the most noticeable reduction in telomere length is taking place in the +BIBR experiment. It would be informative to see how these patterns look like at day 14 for proper comparison to the TEN1 negative cells.

3. In Figure 5, the authors should be careful not to over-interpret their binding data. For the Tel-36 and Non-Tel-48 substrates, they have not effectively measured $K_d(\text{app})$ since binding is saturated at the first point of the titration. The authors should consider doing a finer titration and then fit those data rather than trying to extract apparent binding constants from the data they present. For example, the binding isotherms (which they should consider plotting on either a semi-log plot or broken x-axis) shown in Fig. S5 are interpreted to demonstrate an enhanced effect of TEN1 in the case of the Tel-36 substrate. However, given the large error bars, and the fact that the binding curves for the CS and CST complexes really only differ for one point, this interpretation seems a bit of a reach. The authors should either revise the wording on page 7 to reflect the limitation of these experiments, or do a more careful binding analysis to see if this effect of TEN1 on Tel-36 stands up.

4. The smFRET experiments presented in Fig. 6 and S6 are of good quality; however, the data analysis and some of the experimental design can be improved. First, the authors state they cannot do the smFRET with a native telomere DNA sequence due to insufficient dynamic range to detect CST binding. Given the magnitude of the FRET change observed for the non-telomeric DNA, I would be curious to see what the data on a telomeric substrate looks like. More importantly, the authors interpret the low FRET state observed in the histograms and traces as the CST or CS-bound state. While this is certainly reasonable, to incisively make this conclusion, a protein titration experiment should be done to demonstrate the dependence of k_{on} on the protein concentration. For a simple bimolecular binding interaction, a linear dependence would be expected. Since the protein-binding signal is not direct in this assay, rather a change in DNA structure is detected, this experiment is crucial to support the authors interpretation. Another conclusion that is put forth in this section of the paper is that since the Cy3 dye is not lost upon addition of protein, the CST complex must not disrupt the duplex region of the DNA substrate. It is conceivable that duplex disruption might be incomplete and therefore not drive dissociation of the donor dye. If the authors wish to make this conclusion, they should perform an experiment where the Cy3 is moved to the duplex, either across from the Cy5 or closer to the surface. If the duplex is not disrupted upon protein binding then one would expect the signal to remain constant in the presence or absence of CST complex. The main conclusion of the single-molecule experiments is that TEN1 stabilizes the binding of CS to the DNA. This claim is supported by two pieces of data, neither of which are completely convincing. First, the authors cite the small difference in the dwell time distribution for the CS and CST complexes. There appears to be no description of how the dwell times were determined (HMM fitting?), nor any presentation of a typical dwell time distribution. Simply presenting the plot in Fig6D together with the selected traces shown in Fig. S6 is not sufficient to convince the reader that there is a significant difference in the stabilities of the CS and CST complexes. The authors should describe in greater detail how this analysis was performed. Similarly the authors make a claim that the percent of traces showing 'partial' short-lived binding events is less with the CST complex compared with the CS complex. Again, how was this metric determined, by eye? What was the threshold to call a FRET change 'partial' versus 'full' binding? If the dwell times are exponentially distributed as one might expect, then one can be easily fooled into thinking there are two different types of binding events in a trace, when in fact they are being sampled from the same exponentially distributed process. A more detailed analysis of these dwell times is warranted and I would suggest using the terms short and long-lived rather than partial and full binding.

Minor Points:

1. In Fig. 1F, are ~30% of chromosomes normally expressing gamma-H2AX as shown in the bar-plot? For readers not used to looking at these types of experiments, perhaps explain why this would be since it seems like a high amount of DNA damage signaling.
2. On page 4, in the sentence that begins with "To assess the effect of TEN1 ..." the word "fluorescent" should be changed to "fluorescence".
3. In the last sentence of the first paragraph in the "TEN1 determines DNA binding stability and dynamics" section on page 8, the word "need", should be changed to "needed".
4. On page 10, in the first paragraph, in the sentence that begins with "This interaction might lead to ..." the word "completion" should be changed to "competition".
5. On page 10, in the third paragraph, in the sentence that begins with "The latter possibility seems particularly ..." the word "is" should be removed after "pol alpha...".

Reviewer #3 (Remarks to the Author):

Feng et al. describe an inducible knockout system in the HCT-116 colon cancer cell line to investigate the role of the CST complex in telomere maintenance. The experiments in this study appear to be executed in a careful and expert manner. Overall, the results are both important and compelling and I would recommend this for publication if some important items can be addressed.

Important items:

1. The title should clarify that the results pertain to a single human cancer cell type; as currently written, it overstates the results to appear as though they are a universal property of the complex. This is important because, as the authors themselves emphasize in the discussion, the precise roles that the subunits of this complex perform varies substantially depending on the context. One suggestion is to add to the title the phrase 'in HCT-116 colon cancer cells' or 'in a human colon cancer cell line'.
2. The TEN1^{-/-} phenotype results in telomere shortening, modest G-overhang elongation in S phase (which can arise from telomerase activity and/or C-strand resection), a delay in C-strand fill-in in late S/G2, and a failure to induce DNA damage response (DDR) at very short telomeres. This latter results is especially striking given the frequency of SFE (Fig S1C, D). The authors conclude "Since telomere shortening in the presence of abundant telomerase activity is characteristic of a deficiency in C-strand fill-in (Fig.S1A), our results again indicated that TEN1 is required for this aspect of CST function." However, telomere shortening in the presence of abundant telomerase is also a hallmark of a failure to recruit telomerase to telomeres. In order to safely draw the conclusion as stated, the authors need to show (or argue why they do not need to show) that telomerase is still recruited to similar levels in these cells, or else the conclusion should be restated to address this.
3. The authors conclude that the failure to induce a DDR in TEN1^{-/-} cells is due to residual, but undetectable, telomeric sequence. However, in the CTC^{-/-} cells (Fig 1E), there are γ H2AX foci at telomeres with low telomere signal that are comparable to low telomere signals in the TEN1^{-/-} cells that lack γ H2AX. How do the authors explain this discrepancy?
4. The authors should also test whether TEN1^{-/-} is epistatic with BIBR1532 treatment with respect to telomere shortening; this is a relatively straightforward experiment since the authors have the

tools for this experiment. In the model the authors are proposing, BIBR1532 should lead to additional telomere shortening in the TEN1^{-/-} cells if telomerase is still operating at these telomere ends. However, another possible result from this experiment would be that no further reduction in telomere length would be observed over 14 days after tamoxifen treatment; such a result would suggest that TEN1 is essential for telomerase recruitment and/or extension of telomeres, and further suggest that the DDR-trigger for telomerase (as recently reported in Lee et al. 2015 and Tong et al. 2015) may somehow be linked with TEN1. On the other hand, if additional telomere shortening is observed, according to the authors, phosphorylation of H2AX should then be observed in the TEN1^{-/-} cells; such results would then indicate that the TEN1^{-/-} telomere shortening phenotype is independent of telomerase recruitment/extension.

5. The authors conclude "Since TEN1 interacts with STN1 but not TPP1, this result also implies loss of TEN1 localization. Thus, CTC1 disruption likely causes removal of the entire CST complex from the telomere." The authors need to test this directly by performing telomere ChIP for FLAG-TEN1 in CTC^{-/-} cells; the authors have both of these tools at their disposal. One possible result from this experiment is that TEN1 localizes to telomeres in the absence of CS, raising the possibility that it performs independent functions.

6. The authors state "ChIP analysis showed that the mutant exhibited much reduced telomere localization relative to WT CTC1 (Fig. 4F). We therefore conclude that CTC1 and STN1 are both necessary for telomere localization." However, to draw this inference conclusively, the authors need to perform telomere ChIP with FLAG-CTC in STN1^{-/-} cells to show that wt CTC exhibits reduced telomere localization in the absence of STN1.

7. The authors state that "Quantification of overhang signals indicated that TEN1^{F/F} cells showed a slight increase in overhang abundance as they transitioned from G1 (0 hr) into mid S-phase (6 hrs)." However, the cell cycle data in the supplement clearly indicate that at '0 hours' after release from the double thymidine block, most of the cells are about a 25% of the way through S phase and are not at all in G1. The authors should modify the text to reflect this.

8. The methods appear to lack some attention to detail. Please correct the following:

In paragraph 1, please:

- state the source of the cells
- report the sgRNA sequences
- report the sequence of the donor ssDNA
- state the source of the PX485 plasmid
- state what vector is used to express the GFP

In the section "Telomere FISH and γ -H2AX Staining" please:

- state the supplier of the secondary of antibodies and the dilutions used
- state the source of the gelatin, preferably with a catalog number
- state whether the Triton was Triton X-100 or some other version
- state the source of the Rnase A

In the ChIP section, please:

- state the reference for the previously described ChIP
- state how long the cells were fixed
- state the source of the protease inhibitors
- state the catalog number for the FLAG M2 antibody used
- state how much RNase A and pteoinase K were used
- state the relevant parameters used for Phosphorimager analysis (also in the EMSA section)

In the section for Westerns:

- was NP-40 used, or the common substitute IGEPAL?
- Also please:

state the mass and dilutions of the antibodies used
state what the milk was dissolved in if it wasn't water

Minor points:

1. In the sentence, "The DNA was separated briefly in agarose gels and hybridized with probe to the telomeric G-strand under non-denaturing conditions (Fig. 2A)." should read "The DNA was separated briefly in agarose gels and hybridized with a probe to the telomeric G-strand under non-denaturing conditions (Fig. 2A)."
2. In the sentence "Likewise, examination of the individual Cy3 and Cy5 signals revealed anti-correlated fluorescence", 'correlate' is misspelled.
3. In the sentence "It is striking that that CS is fully capable of limiting telomerase action despite loss of C-strand fill-in", the word 'that' appears consecutively.
4. In Fig S3, to what do the masses above the image refer?

Response to Reviewers' comments:

Reviewer #3 (Remarks to the Author):

Feng et al. describe an inducible knockout system in the HCT-116 colon cancer cell line to investigate the role of the CST complex in telomere maintenance. The experiments in this study appear to be executed in a careful and expert manner. Overall, the results are both important and compelling and I would recommend this for publication if some important items can be addressed.

Important items:

1. The title should clarify that the results pertain to a single human cancer cell type; as currently written, it overstates the results to appear as though they are a universal property of the complex. This is important because, as the authors themselves emphasize in the discussion, the precise roles that the subunits of this complex perform varies substantially depending on the context. One suggestion is to add to the title the phrase 'in HCT-116 colon cancer cells' or 'in a human colon cancer cell line'.

Due to the journal word limits for the title, we have just added "in cancer cells" to the title. However, we have further clarified the scope of our conclusions by also adding "in a human colon cancer cell line" to line 6 of the introduction.

2. The TEN1^{-/-} phenotype results in telomere shortening, modest G-overhang elongation in S phase (which can arise from telomerase activity and/or C-strand resection), a delay in C-strand fill-in in late S/G2, and a failure to induce DNA damage response (DDR) at very short telomeres. This latter results is especially striking given the frequency of SFE (Fig S1C, D). The authors conclude "Since telomere shortening in the presence of abundant telomerase activity is characteristic of a deficiency in C-strand fill-in (Fig.S1A), our results again indicated that TEN1 is required for this aspect of CST function." However, telomere shortening in the presence of abundant telomerase is also a hallmark of a failure to recruit telomerase to telomeres. In order to safely draw the conclusion as stated, the authors need to show (or argue why they do not need to show) that telomerase is still recruited to similar levels in these cells, or else the conclusion should be restated to address this.

We have modified the text on page 6, lines 6-7 to say "This telomere shortening in the presence of abundant telomerase activity fits with TEN1 being required for C-strand fill-in (Fig.S1A)". Thanks for pointing out this unintended problem with our wording. We did not mean to imply that loss of C-strand fill-in is the only cause of telomere shortening.

The reviewer raised an excellent point concerning whether TEN1 might be needed for telomerase recruitment or engagement at the telomere (see also point 4 below). As none of our previous work addressed this important question, we performed a new series of experiments to determine if telomerase could act on telomeres in TEN1^{-/-} cells. These

experiments are described on page 6 in a new section titled “TEN1 is unnecessary for telomerase recruitment” The results are shown in the new figures Fig. 2E-F and Fig. S2G-H.

To address the question, we generated TEN1 conditional cells that overexpress telomerase (hTERT) and looked for additional G-overhang elongation after TEN1 deletion. Telomerase overexpression normally leads to telomere growth. But in the absence of C-strand fill-in, extra telomerase activity can only lead to elongation of the G-strand (the elongated G-strand can't be converted to dsDNA). If TEN1 is needed for telomerase to extend the overhang, then G-strand/G-overhang length in TEN1^{-/-} cells should not change after hTERT overexpression. However, if telomerase can extend the telomere in the absence of TEN1, G-overhang length should increase in the hTERT overexpressing TEN1^{-/-} cells.

What we observed is that G-overhang length increases substantially in TEN1^{-/-} cells that overexpress hTERT. (Fig. 2E-F). We therefore conclude that TEN1 is not needed for telomerase recruitment or telomere elongation.

3. The authors conclude that the failure to induce a DDR in TEN1^{-/-} cells is due to residual, but undetectable, telomeric sequence. However, in the CTC^{-/-} cells (Fig 1E), there are □H2AX foci at telomeres with low telomere signal that are comparable to low telomere signals in the TEN1^{-/-} cells that lack □H2AX. How do the authors explain this discrepancy?

In CTC1^{-/-} cells, the DDR is triggered by the extremely long G-overhangs rather than loss of telomeric dsDNA (see Feng et al. NAR 2017¹). As a result, the DDR does not correlate with the strength of the telomere FISH signals/the length of the telomere dsDNA. Since TEN1^{-/-} cells do not accumulate extremely long G-overhangs, it makes sense that telomeres with detectable telomere sequence (strong or weak FISH staining) fail to trigger a DDR.

We have rewritten the text relating to Fig. 1E to clarify this point (page 4, lines 19-22)

4. The authors should also test whether TEN1^{-/-} is epistatic with BIBR1532 treatment with respect to telomere shortening; this is a relatively straightforward experiment since the authors have the tools for this experiment. In the model the authors are proposing, BIBR1532 should lead to additional telomere shortening in the TEN^{-/-} cells if telomerase is still operating at these telomere ends. However, another possible result from this experiment would be that no further reduction in telomere length would be observed over 14 days after tamoxifen treatment; such a result would suggest that TEN1 is essential for telomerase recruitment and/or extension of telomeres, and further suggest that the DDR-trigger for telomerase (as recently reported in Lee et al. 2015 and Tong et al. 2015) may somehow be linked with TEN1. On the other hand, if additional telomere shortening is observed, according to the authors, phosphorylation of H2AX should then be observed in

the TEN1^{-/-} cells; such results would then indicate that the TEN1^{-/-} telomere shortening phenotype is independent of telomerase recruitment/extension.

As described above, we used the hTERT overexpression approach to address whether TEN1 is essential for telomerase recruitment and/or extension of telomeres. We chose not to perform the BIBR experiment because G-strand extension by telomerase and C-strand fill-in by DNA polymerase are both required to prevent telomere shortening (please see Fig. S1A and new Fig. S2H). As telomerase only extends only the G-strand, C-strand fill-in is absolutely required to elongate the length of the telomere duplex: i.e. they both work in the same pathway and are epistatic. Consequently, inhibition of telomerase and loss of C-strand fill-in will have equal effects on telomere length maintenance (i.e. loss of telomeric dsDNA) and the combined inhibition of telomerase with loss of C-strand fill-in will not be additive

5. The authors conclude “Since TEN1 interacts with STN1 but not TPP1, this result also implies loss of TEN1 localization. Thus, CTC1 disruption likely causes removal of the entire CST complex from the telomere.” The authors need to test this directly by performing telomere ChIP for FLAG-TEN1 in CTC1^{-/-} cells; the authors have both of these tools at their disposal. One possible result from this experiment is that TEN1 localizes to telomeres in the absence of CS, raising the possibility that it performs independent functions.

We have removed the sentence from the text (page 8, lines 2-3) because we do not have CTC1 conditional cells that express FLAG-TEN1 to perform this experiment

We did in fact generate the necessary cell line so that we could answer the reviewer’s point. However, our tissue culture hood broke and the cells got contaminated. Due to time constraints, it is now too late to re-make the cells.

6. The authors state “ChIP analysis showed that the mutant exhibited much reduced telomere localization relative to WT CTC1 (Fig. 4F). We therefore conclude that CTC1 and STN1 are both necessary for telomere localization.” However, to draw this inference conclusively, the authors need to perform telomere ChIP with FLAG-CTC in STN1^{-/-} cells to show that wt CTC exhibits reduces telomere localization in the absence of STN1.

As we don’t have a STN1^{-/-} cell line to do the definitive experiment, we have softened the statement to say that “our data imply that both CTC1 and STN1 are necessary for CST to localize to telomeres” (page 8, line 9).

7. The authors state that “Quantification of overhang signals indicated that TEN1^{F/F} cells showed a slight increase in overhang abundance as they transitioned from G1 (0 hr) into mid S-phase (6 hrs).” However, the cell cycle data in the supplement clearly indicate that at ‘0 hours’ after release from the double thymidine block, most of the cells are about a 25% of

the way through S phase and are not at all in G1. The authors should modify the text to reflect this.

This is a very good point which we overlooked. As observed by the reviewer, and unlike the HeLa cells which we used previously, the double thymidine block leaves the HCT116 TEN1 conditional (TEN1^{F/F} and TEN1^{-/-}) cells in early S rather than at the G1-S boundary. A survey of the literature indicates that this early S block is quite common for HCT116 cells.

The early S block explains the modest increase in G-overhang length when the cells are released into fresh media and allowed to transition into late S. This is because replication and overhang elongation should be partially complete at the time of release. The early S block also explains why the overhangs in the TEN1^{F/F} cells are shorter at 12 hrs after release than at the time of release (0 hrs); 12 hrs is when the most cells are in G1 and so the overhangs have not yet been elongated by telomerase.

We have now corrected the text and explained this point (page 5 lines 25-28).

8. The methods appear to lack some attention to detail. Please correct the following:

The methods section has been expanded and the requested corrections have been made.

In paragraph 1, please:

state the source of the cells

report the sgRNA sequences

report the sequence of the donor ssDNA

state the source of the PX485 plasmid

state what vector is used to express the GFP

In the section "Telomere FISH and γ -H2AX Staining" please:

state the supplier of the secondary antibodies and the dilutions used

state the source of the gelatin, preferably with a catalog number

state whether the Triton was Triton X-100 or some other version

state the source of the Rnase A

In the CHIP section, please:

state the reference for the previously described CHIP

state how long the cells were fixed

state the source of the protease inhibitors

state the catalog number for the FLAG M2 antibody used

state how much RNase A and pteinase K were used

state the relevant parameters used for Phosphorimager analysis (also in the EMSA section)

In the section for Westerns:
was NP-40 used, or the common substitute IGEPAL?
Also please:
state the mass and dilutions of the antibodies used
state what the milk was dissolved in if it wasn't water

Minor points:

1. In the sentence, "The DNA was separated briefly in agarose gels and hybridized with probe to the telomeric G-strand under non-denaturing conditions (Fig. 2A)." should read "The DNA was separated briefly in agarose gels and hybridized with a probe to the telomeric G-strand under non-denaturing conditions (Fig. 2A)."

Corrected

2. In the sentence "Likewise, examination of the individual Cy3 and Cy5 signals revealed anti-corrolated fluorescence", 'correlate' is misspelled.

Corrected

3. In the sentence "It is striking that that CS is fully capable of limiting telomerase action despite loss of C-strand fill-in", the word 'that' appears consecutively.

Corrected

4. In Fig S3, to what do the masses above the image refer?

Average TFU. We have added the TFU

Reviewer #2 (Remarks to the Author):

The human CST complex (CTC1-STN1-TEN1) is an important regulator of telomere homeostasis, with implicated roles in both regulation of telomere extension by the ribonucleoprotein reverse transcriptase telomerase as well as in orchestrating recruitment of the necessary replication proteins required for promoting C-strand fill-in following telomerase action. Although CST has been show to be critical for telomere regulation in a variety of model systems, the precise division of labor between the components of the CST complex in mediating its various cellular activities has not been firmly established. Feng et al. present a detailed study of the human CST complex, using an impressive combination of cell-based assays, together with biochemical and biophysical analyses. The major conclusions of the work are that TEN1 is not required to limit telomerase activity, but is

essential for C-strand fill-in synthesis. In the absence of TEN1, HCT116 cells retain the ability to continue to divide and do not elicit a DNA damage response as measured by gammaH2AX foci (unlike CTC1 negative cells), but do cause a comparable increase in the number of signal free ends and chromosome fusions as CTC1 negative cells. The distinct phenotypes of the TEN1 negative and CTC1 negative cell lines suggested these components of the CST complex serve non-overlapping roles and prompted further investigation by the authors. Using the same cell lines, the authors measured the change in G-strand overhang signal and find that removal of CTC1 has a significantly larger positive effect on the length of the G-strand overhang than that observed for the TEN1 negative cells. This result leads to the conclusion that TEN1 is not required to limit the action of telomerase at telomeres. The authors next turn to telomere length analysis and find that TEN1 removal does cause detectable telomere shortening, but not a decrease in telomerase activity, suggesting the effect of TEN1 removal is to limit C-strand fill-in. The finding that that the CS complex was sufficient to limit telomerase activity prompted subsequent experiments using ChIP to test whether both CTC1 and STN1 were required for telomere localization. The results of the ChIP experiments support the notion that both CTC1 and STN1 are required for wild type levels of telomere localization. Having investigated the properties of CST in their cell lines, the authors take the commendable step to characterize the DNA binding properties of CST and its sub-complexes in vitro. The authors first turn to EMSA experiments to test the binding of CST, CS, and ST to various model telomere DNA substrates. The results of the EMSA analysis suggest that the presence of TEN1 stabilizes the protein-DNA complex as compared with the CS complex alone. Finally, the authors present yet another complementary experiment, using single-molecule FRET to analyze the binding dynamics of the various recombinant CST, CS and ST complexes. The authors observe smFRET signals that they interpret to represent short-lived and long-lived protein bound states, with the presence of TEN1 supporting more long-lived binding events. Overall, the manuscript is clearly written, the experiments well designed, and with a few exceptions (see below) the conclusions are largely supported by the experimental data. Therefore, if the authors can address the specific points below in a revised manuscript, I expect this study will be appropriate for publication in Nature Communications.

Major Points:

1. There are several areas where a description of certain methods should be better explained in the paper. This is true for the TRAP assay results shown in Fig. 3A - how was the assay done and what is being quantified in this bar plot? Also, with respect to the purification of the CST complex, there appears to be no description of how these complexes were made in the methods section. The only details appear to be included in the figure legend of Fig. S5 where it is stated that the proteins were coexpressed in 293T cells and purified by FLAG pull down. More details for the production of the recombinant complexes should be included given how many experiments are done with them. On this

same topic, do the authors have any estimates of the stoichiometry of their complexes? The TEN1 protein appears to be limiting in their gel, but is this just a consequence of weak staining due to its small size. Better characterization and description of the protein complexes would be appreciated and improve the paper.

We have added sections to the materials and methods describing (i) the TRAP assay protocol and how the results were quantified and (ii) expression and purification of recombinant CST or CS complex.

We do not know the stoichiometry of our CST complexes because we have never generated complexes in which all three subunits have the same tag (necessary to accurately determine stoichiometry by western blot). The stoichiometry of the endogenous (human/mamalian) complex also remains to be determined. However, X-ray crystallography of hSTN1-TEN1 complexes revealed a 1:1 stoichiometry so we think the faint staining of TEN1 relative to STN1 most likely reflects the small size of TEN1.

2. In Fig. 3C, the authors should consider showing the same experiment with CTC1^{F/F} cells for proper comparison, since BIBR treatment should result in telomere shortening regardless of CTC1 knockdown. Also, why is the 14-day time point excluded? It seems that day 10 is where the G-strand elongation is starting to become noticeable in the –BIBR experiment, and likewise, at day 10 is when the most noticeable reduction in telomere length is taking place in the +BIBR experiment. It would be informative to see how these patterns look like at day 14 for proper comparison to the TEN1 negative cells.

We apologize to the reviewers for the confusion caused by the labeling of the cell lines in our original Figures 2, 3, S1, S3. All samples labeled as Day 0 of tamoxifen treatment are the TEN1^{F/F} or CTC1^{F/F} control cells. We have changed the labeling in all our figures to make it more apparent that day 0 of tamoxifen treatment always corresponds to CTC1^{F/F} cells (we now use the label CTC1^{Cond.} instead of CTC1^{-/-}).

Related to Fig. 3C, we have added a panel showing an experimental time line (Fig. S3E) to clarify the experimental design. As mentioned above, the day 0 sample corresponds to CTC^{F/F} cells that were treated +/-BIBR for 14 days. The reviewer will notice that the Day 0/CTC1^{F/F} cells treated with BIBR for 14 days had shorter telomeres than their untreated counterparts. Thus, the BIBR did indeed cause telomere shortening regardless of CTC1 status.

We did not include the day 14 time point in Fig. 3C because there were not enough BIBR-treated CTC1^{-/-} cells to do both the telomere length (Fig. 3C) and G-overhang analysis (Fig. 3F-G). The proliferation rate of CTC1^{-/-} cells normally starts to decline after 7-10 days of tamoxifen treatment. The BIBR treatment speeded up the decline, making it hard to get enough cells to work with after 14 days of tamoxifen treatment. We did not think it necessary to do the TRF analysis at day 14 (and hence did not scale up the size of our cultures) because the TRF elongation is most visible in non-BIBR treated CTC1^{-/-} cells at day 10 (Fig. 3C and Feng et al. NAR 2017, 45, 4281-93). It becomes less apparent by day

14 (Feng et al. NAR 2017, 45, 4281-93). Moreover, the BIBR-treated CTC1^{-/-} cells clearly show the lack of elongation and increased telomere shortening by day 10.

3. In Figure 5, the authors should be careful not to over-interpret their binding data. For the Tel-36 and Non-Tel-48 substrates, they have not effectively measured K_d(app) since binding is saturated at the first point of the titration. The authors should consider doing a finer titration and then fit those data rather than trying to extract apparent binding constants from the data they present. For example, the binding isotherms (which they should consider plotting on either a semi-log plot or broken x-axis) shown in Fig. S5 are interpreted to demonstrate an enhanced effect of TEN1 in the case of the Tel-36 substrate. However, given the large error bars, and the fact that the binding curves for the CS and CST complexes really only differ for one point, this interpretation seems a bit of a reach. The authors should either revise the wording on page 7 to reflect the limitation of these experiments, or do a more careful binding analysis to see if this effect of TEN1 on Tel-36 stands up.

We apologize for the confusion. The data for the K_d(app) calculations were not obtained from the EMSAs shown in Fig. 5 as the range of concentrations was indeed inadequate. We performed additional EMSAs with a much wider range of protein concentrations well below the level that saturated binding. We now show examples of these titrations in Fig. S5C-F and refer to them in the text (Page 8, lines 20-21). As requested we have replotted the binding isotherms with a log-scale on the X-axis to better show the lower concentration data points.

4. The smFRET experiments presented in Fig. 6 and S6 are of good quality; however, the data analysis and some of the experimental design can be improved. First, the authors state they cannot do the smFRET with a native telomere DNA sequence due to insufficient dynamic range to detect CST binding.

Given the magnitude of the FRET change observed for the non-telomeric DNA, I would be curious to see what the data on a telomeric substrate looks like.

We have included a figure at the bottom of this response showing the FRET signal obtained with the 10 nt telomeric overhang substrate +/- CST (Fig. R1 below). We confirmed that the same protein preparation was fully active in an EMSA assay using the equivalent EMSA substrate (i.e. a 10 nt ³²P-labeled telomeric overhang substrate). The reviewer will notice that addition of CST does not cause the appearance of an intermediate FRET signal as occurs after CST addition to a substrate with an 18 nt overhang. Since the 18 nt overhang substrate gives a signal of ~0.2 after CST binding (see Fig. 6B), the much shorter 10 nt overhang should give a FRET signal >0.2 when fully extended (i.e. there should be more residual FRET between the donor and acceptor). Since only the 0.9 FRET signal is visible after CST addition, we think that the fully extended 10 nt ssDNA is too short

to prevent efficient FRET. An alternative explanation is that the fluorophore at 5' end of the DNA duplex interferes with CS/CST binding. Either way, experiments with this substrate are uninformative.

More importantly, the authors interpret the low FRET state observed in the histograms and traces as the CST or CS-bound state. While this is certainly reasonable, to incisively make this conclusion, a protein titration experiment should be done to demonstrate the dependence of k_{on} on the protein concentration. For a simple bimolecular binding interaction, a linear dependence would be expected. Since the protein-binding signal is not direct in this assay, rather a change in DNA structure is detected, this experiment is crucial to support the authors interpretation.

We have performed the suggested titration: Different concentrations of CS or CST were added to the slide and imaging was performed in the presence of protein. As now shown in Fig S6A, the fraction of DNA substrate molecules exhibiting high FRET signals depends on protein concentration. These data demonstrate that the change in DNA structure depends on protein binding. We note that the decrease in FRET in response to increased concentrations of CST is not linear, most likely due to the complexity of CST binding. Like RPA, CST binds DNA via multiple OB folds and, as with RPA, binding of the individual OB folds (and the entire complex) is very dynamic². As a result, CST undergoes facilitated displacement where a molecule of unbound CST can cause a molecule of bound CST to dissociate from the DNA (see Bhattacharjee et al. NAR 2017²). The amount of facilitated displacement increases with CST concentration. Consequently we would not expect to see complete loss of FRET/full CST binding at the 5 nM concentration.

To further address the reviewer's concern, we have added an additional figure to provide evidence that the loss of FRET is caused by CST binding rather than photobleaching (Fig S6G). In this experiment, the slide was illuminated with the green laser (532 nm light) for the first 20 sec, capturing two CST binding events and one CST dissociation event. The green laser was then turned off and the red laser (640 nm light) turned on to demonstrate that the Cy5 could still be excited by direct illumination. The switch in illumination was repeated several times to demonstrate that each dye remained active.

Additional support for the assumption that the low FRET state reflects CST/CS binding is provided by our smFRET experiments with ST. Here we do not observe a change in FRET, indicating that the change in FRET depends on protein binding rather than buffer conditions.

Another conclusion that is put forth in this section of the paper is that since the Cy3 dye is not lost upon addition of protein, the CST complex must not disrupt the duplex region of the DNA substrate. It is conceivable that duplex disruption might be incomplete and therefore not drive dissociation of the donor dye. If the authors wish to make this conclusion, they should perform an experiment where the Cy3 is moved to the duplex, either across from the

Cy5 or closer to the surface. If the duplex is not disrupted upon protein binding then one would expect the signal to remain constant in the presence or absence of CST complex.

We agree with the reviewer that partial melting of the duplex would not be detected by the smFRET analysis. Our experiments demonstrate that CS and CST lack the extensive helix destabilizing activity exhibited for RPA, but we cannot rule out destabilization of a few base pairs. We made this point in our previous NAR paper (Bhattacharjee et al. NAR 2017²) and we have now added it to the current manuscript (page 9, 5 lines from the bottom).

We have also added another panel to Fig S5B showing that CS binds very poorly to a non-telomeric ssDNA substrate of 26 nt. The fully annealed FRET substrate has 18 bp duplex and 18 nt ssDNA so partially melted FRET substrate would be unlikely to have <12 nt dsDNA at room temperature (i.e. >24 nt ssDNA). Thus, the new panel showing poor binding to the 26 nt ssDNA supports our conclusion that CS must retain junction recognition activity to be able to bind efficiently to the FRET substrate. We refer to this panel on page 9, last two lines.

We do not think it necessary to perform additional FRET experiments to examine whether CST (or CS) has strand-melting activity as we used multiple approaches to rule out such an activity for CST in our previous NAR paper. Here the focus is on whether or not CS retains the capacity to bind a ss-dsDNA junction.

The main conclusion of the single-molecule experiments is that TEN1 stabilizes the binding of CS to the DNA. This claim is supported by two pieces of data, neither of which are completely convincing. First, the authors cite the small difference in the dwell time distribution for the CS and CST complexes. There appears to be no description of how the dwell times were determined (HMM fitting?), nor any presentation of a typical dwell time distribution. Simply presenting the plot in Fig6D together with the selected traces shown in Fig. S6 is not sufficient to convince the reader that there is a significant difference in the stabilities of the CS and CST complexes. The authors should describe in greater detail how this analysis was performed.

We apologize for not describing how the dwell times were determined. This was an oversight. We have now included an additional figure to illustrate the approach (Fig. S7A) and a description of the approach is included in the figure legend. We have also changed how we depict the dwell time distribution in the main Fig. 6D. Instead of the whisker plot (which we have moved to Fig. S7B) we present a histogram showing the dwell time distribution (number of binding and dissociation events per 4 second time interval).

Similarly the authors make a claim that the percent of traces showing 'partial' short-lived binding events is less with the CST complex compared with the CS complex. Again, how was this metric determined, by eye? What was the threshold to call a FRET change 'partial' versus 'full' binding? If the dwell times are exponentially distributed as one might expect, then one can be easily fooled into thinking there are two different types of binding events in

a trace, when in fact they are being sampled from the same exponentially distributed process. A more detailed analysis of these dwell times is warranted and I would suggest using the terms short and long-lived rather than partial and full binding.

We have now explained more fully how we distinguished between the two types of binding event (based on minimum FRET signal, see below) and point out the short lived nature of the smaller FRET changes (Main text page 10, lines 5-8 and legend to Fig. S7C-D). We have also included a table showing the relative frequency of the long-lived versus short-lived binding events for CS (Fig. S7D).

The full/longer lived binding events that were used to determine dwell times involved a sharp FRET transition from 0.75 to ~0.15 during binding and sharp transition back to 0.75 upon dissociation. The short-lived/partial binding events are clearly different as they involved a FRET transition from 0.75 to 0.4-0.5 (rather than 0.15) before returning to 0.75. Although the scoring of the second class of FRET transitions was based on the partial decrease in FRET signal (to 0.4-0.5 rather than to 0.15), an additional feature was their short duration. The whole binding and release event always occurred within 3-4 seconds.

Our previous studies (Bhattacharjee et al. NAR 2017) demonstrated that a FRET transition from 0.7-0.8 to 0.1-0.2 (the exact figures depend on the DNA substrate) represents full unfolding of a DNA substrate (specifically G-quadruplex unfolding). We suggest that the transition to 0.4-0.5 represents partial CST binding because the DNA is clearly not fully unfolded/linear as might be expected if it was engaged by all the OB-folds involved in CST binding.

Minor Points:

1. In Fig. 1F, are ~30% of chromosomes normally expressing gamma-H2AX as shown in the bar-plot? For readers not used to looking at these types of experiments, perhaps explain why this would be since it seems like a high amount of DNA damage signaling.

HCT116 cells are a human colon cancer cell line with a deficiency in mismatch repair. Although they are largely diploid (the reason they are often used to make gene knockouts), like other cancer cell lines they do exhibit a fairly high level of DNA damage signaling. This may be due to oncogene-induced replication stress.

2. On page 4, in the sentence that begins with “To assess the effect of TEN1 ...” the word “fluorescent” should be changed to “fluorescence”.

Correction made

3. In the last sentence of the first paragraph in the “TEN1 determines DNA binding stability and dynamics” section on page 8, the word “need”, should be changed to “needed”.

Correction made

4. On page 10, in the first paragraph, in the sentence that begins with “This interaction might lead to ...” the word “completion” should be changed to “competition”.

Correction made

5. On page 10, in the third paragraph, in the sentence that begins with “The latter possibility seems particularly ...” the word “is” should be removed after “pol alpha...”.

Correction made

Reviewer #1 (Remarks to the Author):

In this manuscript, authors attempt to dissect the contributions individual human CST subunits play in telomere length maintenance and G-overhang regulation. Using CRISPR/CAS9, they introduced loxP sites to CTC1 or TEN1 loci in HCT116 cells to permit conditional knockout of these genes. As previously documented in several of their previous papers, including one published in NAR (2017), deletion of CTC1 results in slight telomere damage, eliciting a DNA damage response that compromises cell growth. In addition, elongation of the 3' G-overhang while C-strand length decreases are also observed in this cell type. The novelty of the current submission is that TEN1^{-/-} cells do not exhibit telomeric DNA damage and cell growth arrest. Removal of TEN1 only leads to a slight increase in G-overhangs and telomere shortening due to failure to properly synthesize the C-strand. The authors conclude that TEN1 is required for C-strand synthesis. Using a series of biochemical and biophysical experiments, they suggest that both CTC1-STN1 are required for telomerase to terminate DNA synthesis on the G-strand while STN1-TEN1 functions to promote C-strand fill-in by DNA polymerase. Unfortunately, the TIN1 data is not very convincing, reducing my enthusiasm for this paper.

1. I always worry when important conclusions in this paper are drawn from only one CRISPR/CAS9 cell line, given the well documented off-target effects that can occur using this method of gene targeting. The authors claim they generated multiple clones but only chose one for analyses-why? They have to show telomere length data from at least one additional Tam-treated CTC1^{F/F} and TEN1^{F/F} cell line, as well as control CTC1^{F/F} and TEN1^{F/F} cell lines not treated with Tam. These controls are especially important since in Figure 3B and 3C, it appears that telomere lengths in CTC1^{-/-} and TEN1^{-/-} cells, even at day 0 of Tam treatment, are already significantly different from each other. Why? Additional controls as indicated above are needed to resolve this issue.

We have expanded the materials and methods to better explain the steps involved in generating the TEN1 and CTC1 conditional cells and the safeguards that were taken to ensure correct gene targeting of the TEN1 gene locus (the details for the CTC1 locus were previously published¹).

We have also provided the Reviewer with a figure (Fig. R2 below) showing examples of the G-overhang and telomere length analyses that were performed with two additional TEN1 conditional clones and the telomere length analysis from two additional CTC1 conditional clones. These separate clones were isolated after insertion of Cre-ER into a single TEN1 or CTC1 cell line harboring the two inserted LoxP sites.

Due to practical considerations (see below), the CTC1 and TEN1 conditional cells originated from a single cell isolate obtained early in the sequence of steps needed to make these cell lines. The clonal nature of the CTC1 and TEN1 conditional cells explains why they have such a different basal telomere length. Individual cells in a culture of HCT116 cells exhibit significant variability in their average telomere length. Thus, clones derived from separate cells would also be expected to differ in telomere length.

Generation of TEN1 and CTC1 conditional cells. The DNA sequence of the introns at the 5' end of both the TEN1 and CTC1 gene loci is quite repetitive. Consequently, the two LoxP sites had to be introduced by separate rounds of gene targeting. For the TEN1 conditional cells, it was also necessary to screen 2-3 dozen clones obtained from individual GFP positive (Cas9 transfected) cells to obtain 3-4 clones with the required LoxP insertion. Following transfection with the Cre-ER expression vector, it was again necessary to screen multiple clones for efficient tamoxifen-induced gene disruption. Given the amount of labor involved, it not feasible to generate multiple fully independent conditional cell lines. Instead, we picked one clone from the each gene targeting event to then use in the next step. This is a standard approach in generating a complex cell line. At each step we performed the controls described below guard against off-target effects.

For the CTC1 knockout, the gene targeting was performed with adeno-associated virus not CRISPR/CAS9 (see our previous publication) and we had to screen hundreds of clones to obtain 3-4 clones with the required LoxP insertion. Thus, generation of fully independent cell lines was again unfeasible. The gene targeting approach used to generate the CTC1 conditional cell line is described in full in our previous peer reviewed publication (Feng et al, 2017, NAR 24, 4281-93) as is the validation and characterization of the cell line.

Steps taken to guard against off-target effects. We took multiple steps to guard against possible off target effects during the genome editing. **First**, after each genome editing step we sequenced not only the TEN1 gene locus to verify correct LoxP insertion but also five additional loci that showed the next highest homology to the individual gRNAs. **Second**, we are not examining a simple gene knockout where there is significant potential for abnormal phenotypes relative to wild type cells due to off target editing. We instead generated conditional cell lines so we could examine the phenotypic changes in response to acute gene disruption. If there were off-target effects they would be present in both the TEN1^{F/F} and TEN1^{-/-} (or CTC1^{F/F} and CTC1^{-/-}) cells and thus would not have been scored. **Third**, we

generated a rescue cell line where an exogenous TEN1 or CTC1 cDNA was introduced into the TEN1/CTC1 conditional cells. Our analysis of the CTC1 rescue cells was published previously¹. The TEN1 rescue cells are shown in the current manuscript. We demonstrated that in each case the exogenous allele can rescue the effects of TEN1/CTC1 deletion (i.e. tamoxifen treatment). Thus, the phenotypes observed in the TEN1^{-/-} or CTC1^{-/-} cells cannot be due to random LoxP insertions causing deletion of other genomic regions. **Fourth**, we did screen multiple independent clones obtained after the Cre-ER integration step to verify effects on telomere and G-overhang length (Fig. R2).

2. The authors show that telomeres lacking CTC1 activate a damage response and recruits γ H2AX. What is the p53 status in these cells? One would assume it has to be p53^{+/+} to elicit the growth arrest seen in the CTC1 null cells, but the authors need to show this data.

The parental HCT116 cell line used to make the CTC1 and TEN1 conditional cells is wild type for p53. HCT116 cells were the original cell line used by Vogelstein to demonstrate that p53 is required for cells to sustain a G2 arrest after DNA damage³. We now mention that the cells are wild type for p53 in the Materials and Methods (Cell culture and generation of TEN1^{-/-} cells, lines 3-4). We also provide evidence (by RT-PCR and Western blot) of the p53 status in Fig. S1C-D.

3. The novelty of this paper is the authors' suggestion that TEN1 regulates C-strand filling by DNA pol alpha. However, the data presented are rather unconvincing. I do not see any telomere overhang length change in TEN1 null cells after double thymidine block (Supp Figures 2C). The authors need to inhibit DNA pol a in this experiment to make these effects more apparent.

We agree that the TEN1^{F/F} and TEN1^{-/-} cells exhibit only a modest increase in overhang elongation after release from the double thymidine block. This is to be expected as, unless one overexpresses telomerase, the increase in overhang length due to telomerase action/C-strand resection is normally quite small⁴⁻⁶. Moreover, as discussed in our response to reviewer #3, the double thymidine block arrested our cells part way through S-phase so some overhang elongation will have already take place prior to release from the block.

The important point is that the overhangs in the TEN1^{-/-} cells exhibit a decrease in the overhang shortening that normally occurs as the cells transition through G2 into the G1 of the next cell cycle. This delay in overhang shortening is a well-accepted characteristic of a deficiency in C-strand fill-in that has been described by multiple labs^{4,7,8}.

Over the years, we and others have rigorously demonstrated that CST regulates C-strand fill-in^{4,6,7,9}. Since, regulation of C-strand fill-in is now a well-established paradigm for CST function, we consider the cell cycle analysis shown in Fig. S2 to be sufficient to confirm this point.

Regarding the inhibition of DNA pol alpha to prevent C-strand fill-in, such experiments have been attempted in the past by adding aphidicolin or Cdk inhibitors to cells in late S (see Dai et al. 2010, EMBO J. ,29,2788-801) but, interpretation of the data is complicated

and somewhat controversial. This is because aphidicolin impacts non-telomeric DNA replication (which can continue into M-phase) and CdK inhibition affects many additional processes.

Also, since TEN1^{-/-} cells have no growth issues, why not monitor them longer over many more passages to examine the status of the C-strand and the G-overhang?

We have added a new 21 day time course to show that the TEN1^{-/-} cells continue to exhibit only modest overhang elongation (Fig. S2A-B).

The three-week time frame for our growth curves was chosen so we could directly compare the effects of TEN1 and CTC1 depletion (CTC1^{-/-} cells show a large decline in growth between days 14-21). Since the effects of TEN1 and CTC1 depletion on telomere structure were readily apparent within two weeks, we saw no need for a more prolonged analysis.

A TEN1^{F/F} control cell line is needed in all of these experiments.

We again apologize for the confusion caused by the original labeling of our cell lines. As mentioned above, all samples labeled as Day 0 of tamoxifen treatment are the TEN1^{F/F} or CTC1^{F/F} controls. We have altered the labeling and figure legends in all the affected figures to clarify this point.

4. In Figure 3C, addition of the telomerase inhibitor BIBR reduces telomere length in CTC1 null cells, likely by inhibiting telomerase. However, as shown in Supp. Fig. 3C, addition of BIBR did not fully inhibit telomerase activity in these cells. If telomerase activity is still present in BIBR treated CTC1 null cells, how did these telomeres shorten?

In the absence of C-strand fill-in telomeres will shorten regardless of whether telomerase is active or inactive (see our response to reviewer #3 points 2 & 4 and new Fig. S2H). This is because telomere length maintenance requires the generation of telomeric dsDNA (please see Fig. S1A). As telomerase only extends only the G-strand, C-strand fill-in absolutely required to maintain the length of the telomere duplex. Since telomerase action and C-strand fill-in function in the same pathway and are epistatic, disruption of either process will lead to shortening of the telomere duplex DNA.

5. Does TEN1 play a direct role in recruiting DNA pol alpha to the C-strand? Supp figure 6F suggest this, but it is necessary to show Pol a recruitment by CHIP in both TEN^{F/F} and TEN^{-/-} cells.

We do not think that TEN1 plays a direct role in recruiting DNA pol alpha to the telomere (see below for the rational). Rather, we think that TEN1 is important for engaging pol alpha for C-strand synthesis. To clarify this point, we have modified the model in Fig. 6F to indicate pol alpha telomere association even in the absence of a full CST complex. Since

DNA pol alpha is already present at the telomere prior to CST/TEN1 action (see below), we would not expect ChIP to show a change in Pol alpha localization in the TEN1 knockout.

As discussed on page 11 paragraph 2, prior work indicates that CST is not needed to recruit Pol alpha to the telomere (when assayed by ChIP, Pol alpha telomere localization did not decrease after STN1 depletion⁶). Consequently, we have no reason to think that TEN1 plays a role in DNA pol alpha recruitment. Based on our binding studies, we instead think the TEN1 somehow helps DNA pol alpha engage with the G-overhang to perform C-strand synthesis. We note that this model fits well with recent data published by Neil Lue's lab¹⁰ concerning the role of STN1 in directing Pol alpha switching from RNA to DNA synthesis.

We included Supp fig 6F (now Fig. S8) because the data relate directly to our discussion of how TEN1 might promote C-strand fill-in (page 11, paragraph 2). The data indicate that TEN1 does **NOT** interact with pol alpha and is **NOT** required for CTC1 and STN1 to interact with pol alpha. We previously published similar data (Bhattacharjee et al, 2016, PLOS Genetics 12, e1006342), but it was buried in the supplementary figures. We therefore thought we should repeat the interaction analysis and include it in the current manuscript. If the reviewers consider the figure to be superfluous, we can delete it.

6. There's a disconnect between Figure 4 and 5/6. Figure 4 shows that STN1 and CTC1 localizes to telomeres robustly in vivo in the absence of TEN1. However, Figure 5/6 show that CS binding to in vitro telomeric substrates in the absence of TEN1 is unstable. So which is it-they can't be both correct?

Telomere localization does not necessarily depend on DNA binding. CST interacts with the shelterin subunit TPP1 through CTC1 and STN1^{8,11,12}. This interaction with TPP1 probably stabilizes CTC1-STN1 association with the telomere in the absence of TEN1. We have clarified this point in the results section (page 10, last two sentences).

Additional notes:

1. In Supp. Fig 1D, signal free end info need info for STN1^{F/F} and CTC1^{F/F} cells too.

Information for the TEN1^{F/F} and CTC1^{F/F} are represented by the day 0 tamoxifen treatment. As in the other figures, we have adjusted our labeling of the cell lines to clarify this point.

Literature citations relevant to response to reviewer's comments

1. Feng, X., Hsu, S.J., Kasbek, C., Chaiken, M. & Price, C.M. CTC1-mediated C-strand fill-in is an essential step in telomere length maintenance. *Nucleic Acids Res* **45**, 4281-4293 (2017).
2. Bhattacharjee, A., Wang, Y., Diao, J., Price, C. Dynamic DNA binding, junction recognition and G4 melting activity underlie the telomeric and genome-wide roles of human CST. *Nucleic Acids Res*, In Press (2017).
3. Bunz, F. et al. Requirement for p53 and p21 to sustain G2 arrest after DNA damage. *Science*

- 282**, 1497-501 (1998).
4. Wang, F. et al. Human CST has independent functions during telomere duplex replication and C-strand fill-in. *Cell Rep* **2**, 1096-103 (2012).
 5. Kasbek, C., Wang, F. & Price, C.M. Human TEN1 maintains telomere integrity and functions in genome-wide replication restart. *J Biol Chem* **288**, 30139-50 (2013).
 6. Huang, C., Dai, X. & Chai, W. Human Stn1 protects telomere integrity by promoting efficient lagging-strand synthesis at telomeres and mediating C-strand fill-in. *Cell Res* **22**, 1681-95 (2012).
 7. Dai, X. et al. Molecular steps of G-overhang generation at human telomeres and its function in chromosome end protection. *EMBO J* **29**, 2788-801 (2010).
 8. Chen, L.Y., Redon, S. & Lingner, J. The human CST complex is a terminator of telomerase activity. *Nature* **488**, 540-4 (2012).
 9. Gu, P. et al. CTC1 deletion results in defective telomere replication, leading to catastrophic telomere loss and stem cell exhaustion. *EMBO J* **31**, 2309-21 (2012).
 10. Ganduri, S. & Lue, N.F. STN1-POLA2 interaction provides a basis for primase-pol alpha stimulation by human STN1. *Nucleic Acids Res* **45**, 9455-9466 (2017).
 11. Wan, M., Qin, J., Songyang, Z. & Liu, D. OB fold-containing protein 1 (OBFC1), a human homolog of yeast Stn1, associates with TPP1 and is implicated in telomere length regulation. *J Biol Chem* **284**, 26725-31 (2009).
 12. Bhattacharjee, A., Stewart, J., Chaiken, M. & Price, C.M. STN1 OB Fold Mutation Alters DNA Binding and Affects Selective Aspects of CST Function. *PLoS Genet* **12**, e1006342 (2016).

Figures for the reviewers

Figure R1. Single molecule FRET analysis of CST binding to telomeric junction substrate with 10 nt overhang. (A) Cartoon showing design of the substrate. Sequence of overhang oligo: TGGCGACGGCAGCGAGGCTTAGGGTTAG-Cy3. Underline indicates region of dsDNA used for slide anchor. (B) FRET histograms generated with DNA substrate alone or after CST addition.

Figure R2. Analysis of multiple clones showing effects of TEN1 or CTC1 gene disruption on telomere length or G-overhang length. Clones were isolated after introduction of Cre-ER into TEN1^{F/F} or CTC1^{F/F} cells. (A) PCR verifying gene disruption in three additional TEN1 conditional clones (TEN1^{F/F} cells expressing Cre-ER) after growth with tamoxifen for 7 days (TEN1^{-/-}). (B) Western blot showing levels of TEN1 in the same cells as (A). Blot was probed with antibody to TEN1 or actinin as a loading control. (C) G-overhang length in TEN1 conditional clones #6 and #10. Rescue: an exogenous TEN1 rescuing allele was introduced in to the TEN1^{F/F} cells. G-overhang abundance was analyzed by in-gel hybridization following various times of tamoxifen (TAM) treatment. Relative G-overhang abundance is shown below each lane. The signal from the TEN1^{F/F} cells (TAM day 0) is set to 1. (D-E) Southern blots showing terminal restriction fragments in two additional TEN1 (D) or CTC1 (E) conditional clones treated with tamoxifen for the indicated times. Rescue: indicates introduction of an exogenous TEN1 rescuing allele. Mean telomere length is indicated below each lane.

A

B

Reviewers' comments:

Reviewer #1 (Remarks to the Author):

The authors have addressed some of my earlier concerns, but not all. They failed to address one of my original questions (question 1).

1. In figure 4, the authors claim that there is no significant difference in CTC1 and STN1 loading on telomere when TEN1 is depleted. However, I see significantly more loading of CTC1 and STN1 to telomeres in the absence of TEN1 (Figure A, B and D). How do you explain this? If TEN1 is not needed to stabilize S-T interaction on telomeres, how do you explain the in vitro DNA binding assays in figure 5 and 6 that show TEN1 stabilizing the CST complex on telomeres?

2. I asked earlier whether TEN1 plays a role in recruiting DNA pol alpha to telomeres. This necessitates experimental ChIP data to show whether DNA Pol alpha localization to telomeres is perturbed in the absence of TEN1. The authors argue that since DNA pol alpha localization to telomeres is not decreased in the absence of STN1, TEN1 is also not required in this process. I don't buy this argument. The authors' own Supp Fig 6F suggests that TEN1 plays a role in DNA Pol alpha's ability to synthesize the C-strand. Ruling in or ruling out a direct interaction between TEN1 and DNA pol alpha is important, and allows one to think mechanistically what is going on here.

3. The authors also must incorporate Reviewer Figure 2 with Figure 1 and place it into the manuscript. The caveats involved in generating their knockout clones must also be included in the MM section, which is not present in the resubmitted manuscript.

Reviewer #2 (Remarks to the Author):

The authors have done a nice job addressing the concerns that were raised during the initial review process and as a result the manuscript is substantially improved. Therefore, I recommend the study be published in its current form.

Reviewer #3 (Remarks to the Author):

I thank the authors for addressing my concerns and suggestions.

Response to Reviewers' comments:

We were pleased that Reviewers 2 and 3 were happy with our revisions as we worked hard to address their concerns. We were also pleased that Reviewer 1 found most of our revisions satisfactory. We hope that the following responses and revisions alleviate his/her remaining concerns.

Modification in response to original comments from Reviewer #3 point 5.

5. The authors conclude “Since TEN1 interacts with STN1 but not TPP1, this result also implies loss of TEN1 localization. Thus, CTC1 disruption likely causes removal of the entire CST complex from the telomere.” The authors need to test this directly by performing telomere ChIP for FLAG-TEN1 in CTC1^{-/-} cells; the authors have both of these tools at their disposal. One possible result from this experiment is that TEN1 localizes to telomeres in the absence of CS, raising the possibility that it performs independent functions.

When we re-submitted the revised manuscript we did not include the ChIP for FLAG-TEN1 in CTC1^{-/-} cells because our newly established cell line had become contaminated. However, we have since re-generated the cell line and performed three repeats of the ChIP experiment. We have added these data to Fig.4. The slot blot is shown as an extra panel to Fig. 4A and the histogram is a new Fig 4E.

The new ChIP data show that TEN1 is indeed lost from telomeres after CTC1 deletion. This result indicates that TEN1 cannot have a CTC1-STN1-independent role at telomeres. Since CTC1 deletion also causes loss of STN1, it also shows that removal of CTC1 causes the loss of the entire CST complex from the telomere.

We have re-revised the text on page 8, lines 2-4 to state this conclusion.

Response to second round of reviews

Reviewer #1 (Remarks to the Author):

The authors have addressed some of my earlier concerns, but not all. They failed to address one of my original questions (question 1).

1. In figure 4, the authors claim that there is no significant difference in CTC1 and STN1 loading on telomere when TEN1 is depleted. However, I see significantly more loading of CTC1 and STN1 to telomeres in the absence of TEN1 (Figure A, B and D). How do you explain this? If TEN1 is not needed to stabilize S-T interaction on telomeres, how do you explain the in vitro DNA binding assays in figure 5 and 6 that show TEN1 stabilizing the CST complex on telomeres?

We have now included images of the membranes and quantification for each ChIP experiment in Fig. S4C-D so the reviewer can see the raw data. As noted by the reviewer, two of the repeats show an increase in CTC1 and STN1 telomere localization after TEN1 knockout. However the third repeat does not. As a result, the increase lacks statistical significance (p=0.137 for increased STN1 telomere localization and p=0.174 for increased CTC1 localization).

In response to the reviewer's comment, we have altered the text on page 7 (last 3 lines) to state: "The analysis revealed that loss of TEN1 did not decrease CTC1 or STN1 telomere association but rather caused a modest (not statistically significant) increase in association (Fig. 4B-C, Fig. S4D)".

Regarding the apparent discrepancy between the stable *in vivo* localization of CTC1-STN1 to the telomere and the decreased *in vitro* binding stability after TEN1 removal, we think that these differences are easily explained by the different conditions *in vivo* versus *in vitro*.

The binding assays shown in figure 5 and 6 are *in vitro* assays performed with DNA and purified CTC1-STN1 or CTC1-STN1-TEN1. These data clearly show that removal of TEN1 destabilizes CTC1-STN1 binding to DNA. However, at the telomere, the shelterin complex is also present. We and others have shown that CTC1 and STN1 bind to the TPP1 subunit of shelterin in the absence of DNA (see references below). As mentioned previously (reviewer 1 point 6), we think that the direct interaction with TPP1 is most likely what stabilizes CTC1-STN1 association with the telomere. We previously amended the text in the results section to clarify this point (the last two sentences of the results section on page 10, now highlighted in red).

References and associated figures showing interaction between TPP1 and CTC1 or STN1:

1. Chen L.Y., Redon, S. & Lingner, J. The human CST complex is a terminator of telomerase activity. *Nature* **488**, 540-4 (2012). **Fig. 3E-F, Fig. S8**
2. Wan, M., Qin, J., Songyang, Z. & Liu, D. OB fold-containing protein 1 (OBFC1), a human homolog of yeast Stn1, associates with TPP1 and is implicated in telomere length regulation. *J Biol Chem* **284**, 26725-31 (2009). **Topic of the entire manuscript**
3. Bhattacharjee, A., Stewart, J., Chaiken, M. & Price, C.M. STN1 OB Fold Mutation Alters DNA Binding and Affects Selective Aspects of CST Function. *PLoS Genet* **12**, e1006342 (2016). **Fig. 4D.**

2. I asked earlier whether TEN1 plays a role in recruiting DNA pol alpha to telomeres. This necessitates experimental ChIP data to show whether DNA Pol alpha localization to telomeres is perturbed in the absence of TEN1. The authors argue that since DNA pol alpha localization to telomeres is not decreased in the absence of STN1, TEN1 is also not required in this process. I don't buy this argument. The authors' own Supp Fig 6F suggests that TEN1 plays a role in DNA Pol alpha's ability to synthesize the C-strand. Ruling in or ruling out a direct interaction between TEN1 and DNA pol alpha is important, and allows one to think mechanistically what is going on

We respectfully disagree with the reviewer concerning the need for pol alpha ChIP data to determine whether TEN1 recruits pol alpha to the telomere. As outlined below, we consider our new ChIP data concerning loss of TEN1 telomere localization after CTC1 disruption (Fig 4A & E), combined with our other data and data published by other labs, together provide convincing evidence that TEN1 is not needed for pol alpha recruitment to telomeres.

Moreover, as explained below, we do not think that ChIP can be used to address how pol alpha is recruited to the G-overhang to perform C-strand fill-in. This is because ChIP cannot reveal the location on the telomere to which pol alpha is recruited.

(Lack of) TEN1 interaction with DNA pol alpha

We are unsure why the reviewer thinks that Figure S6F (initial submission, now Fig. S8) demonstrates an interaction between TEN1 and pol alpha. We have added an arrow to the figure to indicate the pol alpha band. This band is clearly absent from the STN1-TEN1 immunoprecipitate. Perhaps the confusion arose because of the background signal at the top of

the membrane slice. This background signal does not correspond to pol alpha. The reviewer can find another image showing the lack of stable interaction between pol alpha and STN1-TEN1 in Fig. S2B of Bhattacharjee et al, 2016, PLOS Genetics 12, e1006342.

Need for pol alpha chip in TEN1^{-/-} cells

Before we discuss the data relating to TEN1 recruitment we need to raise several relevant points: First, it is important to distinguish between “recruitment” of pol alpha to the telomere versus “engagement” of pol alpha to perform DNA synthesis. We most certainly think, and have clearly demonstrated, that TEN1 is essential to “engage” DNA pol alpha for C-strand fill-in. However, the results listed below provide evidence that TEN1 is not needed to “recruit” pol alpha to the telomere.

Second: It is important to note that pol alpha should localize to both the telomere duplex and the G-overhang during telomere replication. As a population of cells will have some telomeres undergoing telomere duplex replication and others exhibiting C-strand fill-in, a ChIP experiment will report on pol alpha in both locations at once. Currently there is no ChIP protocol that allows one to specifically look at pol alpha (or any other protein) localization to the G-overhang. Consequently, it is not possible to truly assay for pol alpha recruitment to the G-overhang (i.e. to perform C-strand fill-in).

Data indicating that TEN1 is not needed to “recruit” pol alpha to the telomere.

- (1) Pol alpha ChIPs to telomeres after STN1 knockdown (Huang et al. *Cell Res* **22**, 1681-95 (2012) and after CTC1 knockout.

We did not mention the CTC1 knockout data in our previous response to reviewers because it is not published. But we have included a reviewer figure (Fig. R1A), showing this data. It is clear from both our data and that of the Chai lab (Fig. R1B) that pol alpha recruitment to telomeres is not decreased by loss of STN1 or CTC1. Rather, pol alpha association is increased (see below for comments on the increase).

- (2) Our new TEN1 ChIP (new Fig. 4A & D) shows that TEN1 is lost from the telomere after CTC1 disruption. STN1 is also lost after CTC1 disruption.

Points 1 & 2 together indicate that pol alpha is still present at the telomere even after severe depletion, or complete loss of all CST subunits including TEN1. Thus, TEN1 cannot be needed for pol alpha recruitment/telomere localization. The apparent lack of direct interaction between pol alpha and TEN1 further supports this contention.

The increase in pol alpha at telomeres after CST depletion

We were initially puzzled by the data from Dr Chai’s lab showing that STN1 knockdown causes increased pol alpha at telomeres. This is why we repeated the pol alpha ChIP in CTC1^{-/-} cells (Fig. R1). However, we now understand these results. Loss of CST slows telomere duplex replication and leads to an increase in replication fork stalling. Since replication fork stalling is known to increase chromatin association of replicative polymerases, it makes sense that loss of CST would increase pol alpha accumulation.

The accumulation of pol alpha on the telomere duplex (due to replication fork stalling) after CST removal means it is not possible to use ChIP to monitor what is happening to pol alpha on the overhang. Even if TEN1 were needed to recruit pol alpha to the G-overhang for C-strand fill-in, any loss of pol alpha from the overhang in TEN1^{-/-} cells would be masked by the increase in pol alpha on the telomere duplex. Thus, ChIP is not an appropriate way to test for changes in recruitment of pol alpha to the G-overhang for C-strand synthesis.

Fig. R1. ChIP performed using Pol alpha antibody using chromatin from *CTC1^{F/F}* or *CTC1^{-/-}* cells. Left: slot blot. Right: Quantification.

3. The authors also must incorporate Reviewer Figure 2 with Figure 1 and place it into the manuscript. The caveats involved in generating their knockout clones must also be included in the MM section, which is not present in the resubmitted manuscript.

We have incorporated the Reviewer Figure 2 into the supplemental figures because it is too large to add onto Figure 1. It is now part of Fig. S2. We refer to the data in the text on page 4, line 2 so it is clearly associated with Fig. 1. We also refer to the data on page 4, line 37 and page 6 lines 5-6 and line 10 to associate the data with Fig. 2 & 34.

We have amended to the materials and methods to explain the caveats and controls involved in generating the knockout clones (Cell culture and generation of *TEN1^{F/F}* cells). The changes are highlighted in red.

Reviewer #2 (Remarks to the Author):

The authors have done a nice job addressing the concerns that were raised during the initial review process and as a result the manuscript is substantially improved. Therefore, I recommend the study be published in its current form.

Reviewer #3 (Remarks to the Author):

I thank the authors for addressing my concerns and suggestions.

REVIEWERS' COMMENTS:

Reviewer #1 (Remarks to the Author):

The authors have adequately addressed my concerns.